# Encoding Weights of Irregular Sparsity for Fixed-to-Fixed Model Compression

**Baeseong Park**[1]*, **Se Jung Kwon**[1]*, **Daehwan Oh**[2], **Byeongwook Kim**[1], **Dongsoo Lee**[1]
[1]NAVER CLOVA,
`{baesung.park,sejung.kwon,byeonguk.kim,dongsoo.lee}@navercorp.com`
[2]Samsung Research, `dhdh.oh@samsung.com`

## Abstract

Even though fine-grained pruning techniques achieve a high compression ratio, conventional sparsity representations (such as CSR) associated with irregular sparsity degrade parallelism significantly. Practical pruning methods, thus, usually lower pruning rates (by structured pruning) to improve parallelism. In this paper, we study fixed-to-fixed (lossless) encoding architecture/algorithm to support fine-grained pruning methods such that sparse neural networks can be stored in a highly regular structure. We first estimate the maximum compression ratio of encoding-based compression using entropy. Then, as an effort to push the compression ratio to the theoretical maximum (by entropy), we propose a sequential fixed-to-fixed encoding scheme. We demonstrate that our proposed compression scheme achieves almost the maximum compression ratio for the Transformer and ResNet-50 pruned by various fine-grained pruning methods.

## 1 Introduction

As one of the efficient compression methods, pruning reduces the number of parameters by replacing model parameters of low importance with zeros (LeCun et al., 1990). Since magnitude-based pruning has shown that pruning can be conducted with low computational complexity (Han et al., 2015), various practical pruning methods have been studied to achieve higher compression ratio (Zhu and Gupta, 2017; Molchanov et al., 2017; Louizos et al., 2018; Gale et al., 2019). Recently, pruning has been extended to a deeper understanding of weight initialization. Based on the *Lottery Winning Ticket* hypothesis (Frankle and Carbin, 2018), (Renda et al., 2020) suggests a weight-rewinding method to explore sub-networks from full-trained models. Furthermore, pruning methods at initialization steps without pre-trained models have also been proposed (Lee et al., 2019b; Wang et al., 2020).

Despite a high compression ratio, fine-grained pruning that eliminates each parameter individually has practical issues to be employed in parallel computing platforms. One of the popular formats to represent sparse matrices after pruning is the Compressed Sparse Row (CSR) whose structures are irregular. For parallel computing, such irregular formats degrade inference performance that is dominated by matrix multiplications (Gale et al., 2020). Algorithm 1 presents a conventional sparse matrix-vector multiplication (SpMV) algorithm using CSR format which involves irregular and data-dependent memory accesses. Correspondingly, performance gain using a sparse matrix multiplication (based on CSR) is a lot smaller than the compression ratio of pruning (Yu et al., 2017). Structured pruning methods (e.g, block-based pruning (Narang et al., 2017; Zhou et al., 2021), filter-based pruning (Li et al., 2017), and channel-based pruning (He et al., 2017; Liu et al., 2017)) have been suggested to enhance parallelism by restricting the locations of pruned weights. Those methods, however, induce degraded accuracy and/or lower pruning rate than fine-grained pruning.

In this paper, as an efficient method to compress sparse NNs pruned by fine-grained pruning, we consider weight encoding techniques. As shown in Algorithm 2, encoded weights are multiplied by a fixed binary matrix $M^\oplus$ to reconstruct the original weights. We propose an encoding method and $M^\oplus$ design methodology to compress sparse weights in a regular format. It should be noted that **a sparse matrix multiplication can be even slower than a dense matrix multiplication unless pruning rate is high enough** (Yu et al., 2017; Gale et al., 2020) that does not happen with Algorithm

---

*Equal contribution.

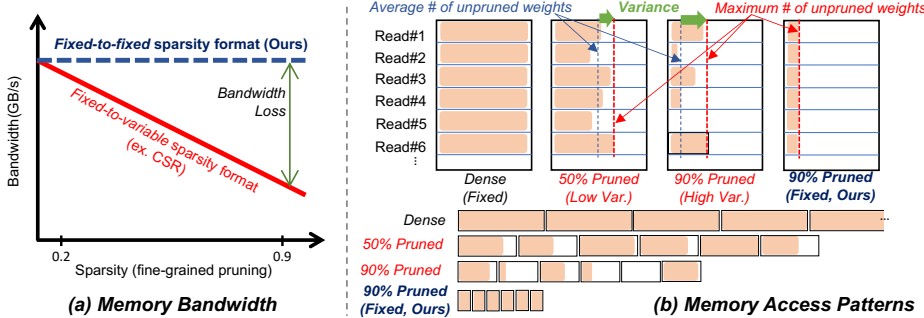

Figure 1: Comparison between fixed-to-variable (e.g., CSR) sparsity format and fixed-to-fixed (proposed) sparsity format. (a): Memory bandwidth comparison. (b): Memory access patterns with irregular sparsity.

2 for memory-intensive workloads. We study the maximum compression ratio of such encoding-based compression using entropy and propose a sequential fixed-to-fixed scheme that keeps high parallelism after fine-grained pruning. We show that by our proposed fixed-to-fixed scheme, a compression ratio can approach the maximum (estimated by entropy) even under the variation of the unpruned weights in a block.

| **Algorithm 1:** SpMV (CSR format) | **Algorithm 2:** Proposed SpMV (using encoded weights) |
|---|---|
| In: Dense vector $\boldsymbol{x}$, 
     CSR vectors $\boldsymbol{dat}, \boldsymbol{row}, \boldsymbol{col}$ 
 Out: Dense vector $\boldsymbol{y}$ 
 **for** $i \leftarrow 1$ **to** $n$ **do** 
     **for** $j \leftarrow \boldsymbol{row}_i$ **to** $\boldsymbol{row}_{i+1}$ **do** 
         $\boldsymbol{y}_i \leftarrow \boldsymbol{y}_i + \boldsymbol{dat}[j] \times \boldsymbol{x}[col[j]]$ 

 /*Irregular, data-dependent access*/ | In: Dense vector $\boldsymbol{x} \in R^m$, Encoded vectors $\boldsymbol{w}_{1..n}^e \in R^k$ 
     Fixed matrix $\boldsymbol{M}^{\oplus} \in \{0,1\}^{k \times m}$, Mask    ** $(k \ll m)$ 
 Out: Dense vector $\boldsymbol{y}$ 
 **for** $i \leftarrow 1$ **to** $n$ **do** 
     $\boldsymbol{W}_i \leftarrow \boldsymbol{w}_i^e \times \boldsymbol{M}^{\oplus}$ over GF(2) 
     $\boldsymbol{y}_i = \boldsymbol{W}_i \cdot \boldsymbol{x}$ with Mask (for zero skipping) 

 /* Decoding $m$ elements using $w_i^e$ (Regular access)*/ |

## 2    FIXED-TO-FIXED SPARSITY ENCODING

Data compression is a process of encoding original data into a smaller size. If a fixed number of original bits are encoded into a fixed number of (smaller) bits, such a case is categorized into a "fixed-to-fixed" compression scheme. Similarly, "fixed-to-variable", "variable-to-fixed", and "variable-to-variable" categories are available while variable lengths of original and/or encoded bits allow higher compression ratio than fixed ones (e.g., Huffman codes (Huffman, 1952) as fixed-to-variable scheme, Lempel-Ziv (LZ)-based coding (Ziv and Lempel, 2006) as variable-to-fixed scheme, and Golomb codes (Golomb, 1966) as variable-to-variable scheme).

Among those 4 categories, "fixed-to-fixed" compression is the best for NNs that rely on parallel computing systems. Fixed-to-fixed compression schemes are, however, challenging when fine-grained pruning is employed in NNs because the number of unpruned weights in a fixed-size block varies. Accordingly, most previous sparsity representations (such as CSR format) translate a fixed-size weight block into a variable-size block while such a translation would demand non-uniform memory accesses that lead to significantly degraded memory bandwidth utilization as shown in Figure 1.

Specifically, in the case of fixed-to-variable sparsity format (e.g., CSR) in Figure 1, we observe that memory bandwidth is low because fine-grained pruning induces a variable number of pruned weights for a certain block or row while memory is designed to access a fixed amount of consecutive data. Since higher sparsity leads to higher relative standard deviation (i.e., coefficient of variation) on pruned weights in a block, low memory bandwidth is a significant issue even though the amount of data to be stored is reduced (see Appendix A). As a result, for fixed-to-variable sparsity format, it is difficult to implement fine-grained pruning with parallel computing systems that require high memory bandwidth (Yu et al., 2017). On the other hand, fixed-to-fixed compression schemes in Figure 1 can maintain the same memory bandwidth regardless of sparsity.

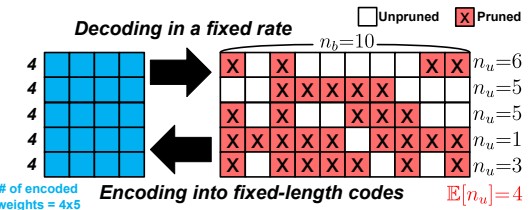

Figure 2: Fixed-to-fixed compression of a sparse weight matrix. Even when a block involves a varying number of unpruned weights, the size of each encoded block is fixed and determined by an average number of unpruned weights in blocks.

| Random Number Generator | | Weight to be encoded | |
|---|---|---|---|
| | | X0X1 | 0X0X |
| 00 → XOR-gates → 0000 | | Unmatched | Matched |
| 01 → XOR-gates → 1110 | | Unmatched | Unmatched |
| 10 → XOR-gates → 0101 | | Unmatched | Matched |
| 11 → XOR-gates → 1001 | | Matched | Unmatched |

Figure 3: Encoding of weights using an XOR-gate decoder as a random number generator.

In this work, we propose a "fixed-to-fixed" compression scheme as shown in Figure 2 when the number of pruned weights in a block can vary. A successful fixed-to-fixed compression of sparse NNs should consider the followings:

- **(High compression ratio)** The maximum compression ratio is limited by the minimum entropy (that can be obtained by a fixed-to-variable scheme as we discuss in Appendix D). Suppose that a block to be encoded contains (fixed) $n_b$ bits among which (fixed) $n_u$ bits are unpruned. A fixed-to-fixed encoding scheme is required to support high compression close to $(n_b/n_u)$ (estimated by entropy). Fixed-to-fixed decoding, then, accepts (fixed) $N_{in}$ bits as an input and produces $N_{out}$ bits as an output while $N_{out}/N_{in} = n_b/n_u$.

- **(Variation tolerance)** For a fine-grained pruning, $n_u$ is given as a random variable whose distribution is affected by pruning rate, $n_b$ size, a particular pruning method, and so on. Our goal is to maintain a fixed-to-fixed scheme with a high compression ratio even under $\mathrm{Var}[n_u] \neq 0$. In Figure 2, for example, 5 blocks of original data have various $n_u$ values while the size of an encoded block is fixed to be 4 ($=\mathbb{E}[n_u]$). We will discuss how to design a variation tolerant encoding.

## 3 RANDOM NUMBER GENERATOR

Before we discuss compression schemes, let us assume that a binary masking matrix is given to represent which weights are pruned or not (note such a binary masking matrix can be compressed significantly (Lee et al., 2019a)). Then, a pruned weight can be described as a don't care value (✗) that is to be masked. We also assume that 1) pruning each weight is performed independently with pruning rate $S$ and 2) unpruned weight is assigned to 0 or 1 with equal probability (such assumptions are not necessary when we demonstrate our experimental results in Section 5).

To obtain both "high compression ratio" and "variation tolerance" while a fixed-to-fixed compression scheme is considered, we adopt random number generator schemes that enable encoding/decoding of weights. A random number generator accepts a fixed number of inputs and produces $n_b$ bits so as to implement a fixed-to-fixed compression scheme. As shown in Figure 3, a weight block is compared with every output of a random number generator. If there is an output vector matching original (partially masked) weights, then a corresponding input vector of a random number generator can be an encoded input vector. As an effort to increase the Hamming distance between any two outputs (i.e., the number of bit positions in which two bits are different), $2^{n_u}$ outputs out of $2^{n_b}$

| $N_{in}$ | $S$=0.5 | 0.6 | 0.7 | 0.8 | 0.9 |
|---|---|---|---|---|---|
| 4 | 90.03 (±7.03) | 89.98 (±6.85) | 89.48 (±7.41) | 90.35 (±3.80) | 90.24 (±2.51) |
| 8 | 94.99 (±2.28) | 95.02 (±1.54) | 95.34 (±1.29) | 95.07 (±0.82) | 95.00 (±0.81) |
| 12 | 96.75 (±0.68) | 96.74 (±0.41) | 97.06 (±0.54) | 96.72 (±0.35) | 96.72 (±0.35) |
| 16 | 97.53 (±0.36) | 97.55 (±0.21) | 97.65 (±0.17) | 97.55 (±0.36) | 97.57 (±0.31) |
| 20 | 98.05 (±0.18) | 98.06 (±0.12) | 98.19 (±0.14) | 98.05 (±0.17) | 98.05 (±0.26) |

(a) $n_u$ (=a number of unpruned weights in a block) is fixed to be $N_{in}$ (i.e., Var$[n_u]$=0).

| $N_{in}$ | $S$=0.5 | 0.6 | 0.7 | 0.8 | 0.9 |
|---|---|---|---|---|---|
| 4 | 88.24 (±7.23) | 88.46 (±6.17) | 88.32 (±5.44) | 87.82 (±4.31) | 87.67 (±2.33) |
| 8 | 94.12 (±1.51) | 93.72 (±1.43) | 93.91 (±1.21) | 93.46 (±1.10) | 93.22 (±0.90) |
| 12 | 95.86 (±0.83) | 95.78 (±0.45) | 95.91 (±0.75) | 95.43 (±0.32) | 95.36 (±0.35) |
| 16 | 96.82 (±0.32) | 96.71 (±0.22) | 96.68 (±0.25) | 96.59 (±0.19) | 96.41 (±0.26) |
| 20 | 97.44 (±0.13) | 97.28 (±0.15) | 97.32 (±0.14) | 97.12 (±0.17) | 97.03 (±0.24) |

(b) $n_u$ follows a binomial distribution $B(N_{out}, 1 - S)$.

| $N_{in}$ | $S$=0.5 | 0.6 | 0.7 | 0.8 | 0.9 |
|---|---|---|---|---|---|
| 4 | 88.31 (±7.19) | 88.07 (±6.09) | 87.16 (±5.49) | 86.91 (±4.39) | 86.67 (±3.03) |
| 8 | 93.75 (±1.67) | 93.19 (±1.38) | 93.00 (±1.03) | 92.61 (±1.01) | 92.39 (±0.60) |
| 12 | 95.59 (±0.67) | 95.23 (±0.72) | 94.89 (±0.50) | 94.71 (±0.33) | 94.50 (±0.28) |
| 16 | 96.43 (±0.36) | 96.18 (±0.25) | 95.69 (±0.13) | 95.73 (±0.12) | 95.56 (±0.16) |
| 20 | 97.08 (±0.12) | 96.79 (±0.09) | 96.32 (±0.09) | 96.40 (±0.08) | 96.35 (±0.13) |

(c) $n_u$ is empirically obtained by pruning the first decoder layer of the Transformer using a magnitude-based pruning method.

Figure 4: Encoding efficiency (%) of random XOR-gate decoders. $S$ is pruning rate and $N_{out}$ is given as $\lfloor N_{in} \cdot (1/(1-S)) \rfloor$.

possible candidates need to be randomly selected. Note that random encoding has already been suggested by Claude Shannon to introduce channel capacity that is the fundamental theory in digital communication (Morelos-Zaragoza, 2006). Since then, practical error correction coding techniques have been proposed to implement random-like coding by taking into account efficient decoding (instead of using a large look-up table).

Similar to error correction coding that usually depends on linear operations over Galois Field with two elements, or $GF(2)$ (Morelos-Zaragoza, 2006), for simple encoding of original data, recently, two compression techniques for sparse NNs have been proposed. An XOR-gate decoder produces (a large number of) binary outputs using (a relatively much smaller number of) binary inputs while outputs are quantized weights (Kwon et al., 2020). Another example is to adopt a Viterbi encoding/decoding scheme (Forney, 1973) to generate multiple bits using a single bit as an input (Ahn et al., 2019). For a block that cannot be encoded into a compressed one by a random number generator, we can attach patch data to fix unmatched bits (Kwon et al., 2020) or re-train the model to improve the accuracy (Ahn et al., 2019).

To compare the random number generation capability of various block-level compression schemes, we introduce 'encoding efficiency' given as a percentage.

$$E \text{ (Encoding Efficiency)} = \frac{\text{\# of correctly matched bits}}{\text{\# of unpruned bits}} \times 100(\%) \qquad (1)$$

Let $S$ be pruning rate ($0 \leq S \leq 1$). To measure encoding efficiency ($E$), we assume that the compression ratio of a random number generator (=the number of output bits / the number of input bits) is $1/(1 - S)$. We generate numerous randomly pruned (binary) weight blocks, and for each block, we investigate all of the possible outputs that a random number generator can produce. If there is a block missing a matching output of a generator, then the maximum number of correctly matched bits is recorded for each block. We repeat such an experiment for all of the blocks. Note that $E$ cannot be higher than 100% for any generators.

### 3.1 FIXED PRUNING RATE IN A BLOCK

For simplicity, we assume that $n_u$ in a block is a fixed number. Let us study $E$ when $n_u$ is fixed using an XOR-gate decoder introduced in (Kwon et al., 2020). For an XOR-gate decoder, when $N_{out}$ is the number of output bits and $N_{in}$ is the number of input bits, a matrix $\boldsymbol{M}^{\oplus} \in \{0, 1\}^{N_{out} \times N_{in}}$ presents connectivity information between an input vector $\boldsymbol{w}^x (\in \{0, 1\}^{N_{in}})$ and an output vector $\boldsymbol{w}^y (\in \{0, 1\}^{N_{out}})$ such that we have $\boldsymbol{w}^y = \boldsymbol{M}^{\oplus} \cdot \boldsymbol{w}^x$ over $GF(2)$. For example, if the second row of $\boldsymbol{M}^{\oplus}$ (with $N_{in} = 4$ and $N_{out} = 8$) is given as $[1\,0\,1\,1]$, then $\boldsymbol{w}_2^y = \boldsymbol{w}_1^x \oplus \boldsymbol{w}_3^x \oplus \boldsymbol{w}_4^x$ ('$\oplus$' indicates a binary XOR operation or an addition over $GF(2)$ equivalently). An element of $\boldsymbol{M}^{\oplus}$ is randomly filled with 0 or 1 as a simple random number generator design technique (Kwon et al., 2020).

To measure $E$, let $N_{in}/N_{out} \approx 1 - S$ such that $N_{out} = \lfloor N_{in} \cdot (1/(1-S)) \rfloor$. Correspondingly, for a certain $S$, a block size (=$N_{out}$) increases as $N_{in}$ increases. When $n_b = N_{out}$ and $n_u = N_{in}$,

Figure 4a describes statistics of $E$ when random $\boldsymbol{M}^{\oplus}$ matrices are associated with random blocks. From Figure 4a, it is clear that increasing $N_{in}$ is the key to improving encoding efficiency. Note that, however, increasing $N_{in}$ and $N_{out}$ complicates the decoding complexity (due to large $\boldsymbol{M}^{\oplus}$) and the encoding complexity as well (due to an exponentially large search space).

## 3.2 VARIABLE PRUNING RATE IN A BLOCK

Now we allow $n_u$ in a block to fluctuate. For Figure 4b, we assume that pruning each weight is a Bernoulli event (with $S$ probability) such that $n_u$ in a block follows a binomial distribution $B(N_{out}, 1-S)$ (thus, $\mathbb{E}[n_u] = N_{out}(1-S)$ and $\mathrm{Var}[n_u] = N_{out}S(1-S)$). $N_{in}$ is given as $\mathbb{E}[n_u]$ in Figure 4b. Compared to Figure 4a, $E$ becomes lower mainly because some blocks would have $n_u$ larger than $N_{in}$ (i.e., too many unpruned weight bits that a random number generator cannot target).

Note that the coefficient of variation of $n_u$ ($=\sqrt{\mathrm{Var}[n_u]}/\mathbb{E}[n_u] = \sqrt{S/(N_{out}(1-S))}$) decreases as $N_{out}$ increases. Indeed, the gap between $E$ in Figure 4a and $E$ in Figure 4b tends to be slightly reduced when $N_{in}$ (as well as corresponding $N_{out}$) increases. To illustrate, as shown in Figure 5, when there are two blocks of different $n_u$, concatenating two blocks into a block (thus, increasing $N_{out}$) can increase the probability of successful encoding due to efficient usage of $N_{in}$. Increasing $N_{in}$ and $N_{out}$ by $n$ times, however, requires an XOR-gate decoder to be larger by $n^2$ times with only a marginal gain in $E$.

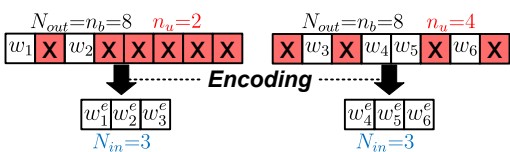

Figure 5: Encoding of two blocks when a number of unpruned weights can vary in a block.

Another way to improve $E$ under the variation on $n_u$ is to decrease $N_{out}$ and increase $N_{in}$. In this case, however, the compression ratio is reduced consequently. We need a solution to design a fixed-to-fixed sparsity compression technique that can improve $E$ of a random number generator under the variation on $n_u$ without sacrificing compression ratio (i.e., $N_{in}/N_{out} = 1-S$).

To validate our observations obtained by synthetic random data, we compute $E$ of an XOR-gate decoder using the Transformer model (Vaswani et al., 2017). The first fully-connected layer of the Transformer is pruned by a magnitude-based pruning method (Han et al., 2015) with pruning rate $S$. Interestingly, $E$ described in Figure 4c is similar to that of Figure 4b. As such, our assumption that pruning a weight is a Bernoulli event is valid for the context of magnitude-based pruning.

## 4 PROPOSED SEQUENTIAL ENCODING SCHEME

If $\mathrm{Var}[n_u]$ is non-zero and $N_{in}$ is fixed, then blocks of small $n_u$ ($< N_{in}$) would have many possible input vectors with matching output vectors while other blocks with large $n_u$ ($> N_{in}$) may have no any single possible input vector. Such unbalance among encoding success rates over blocks can be mitigated if a part of input vectors associated with a block of small $n_u$ can be reused for the neighboring blocks of large $n_u$. In other words, by sharing some parts of an input vector for multiple consecutive blocks (of diverse $n_u$ values), input search space size of each block can be balanced. Reusing inputs is fulfilled by shift registers that have been also introduced to convolutional codes such as Viterbi codes (Morelos-Zaragoza, 2006). In this section, we propose sequential encoding techniques to address the limitations of previous fixed-to-fixed compression schemes (for example, XOR-gate-only decoder (Kwon et al., 2020) lacking tolerance for $n_u$ variation, and Viterbi encoders (Ahn et al., 2019) that receive only one bit as an input (i.e., $N_{in}$ is restricted to be 1).

**Weight manipulation** Since our encoding/decoding techniques process data in a block-level (in the form of a 1-D vector), original sparse weight matrices (or tensors) need to be reshaped through grouping, flattening, and slicing. Assuming that a number format has the precision of $n_w$ bits (e.g., $n_w$=32 for FP32), as the first step of weight manipulation, a weight matrix $\boldsymbol{W} \in \mathbb{R}^{m \times n}$ is grouped into $n_w$ binary matrices $\boldsymbol{W}^b_{1..n_w} \in \{0,1\}^{m \times n}$ (otherwise, $n_w$ successive bits are pruned or unpruned). Then, each binary matrix (or tensor) $\boldsymbol{W}^b_i$ is flattened to be a 1-D vector and each vector is sliced into $\boldsymbol{w}^b_{i,1..l}$ blocks when $l$ ($=\lceil \frac{mn}{N_{out}} \rceil$) indicates the number of blocks in a 1-D (flattened) vector.

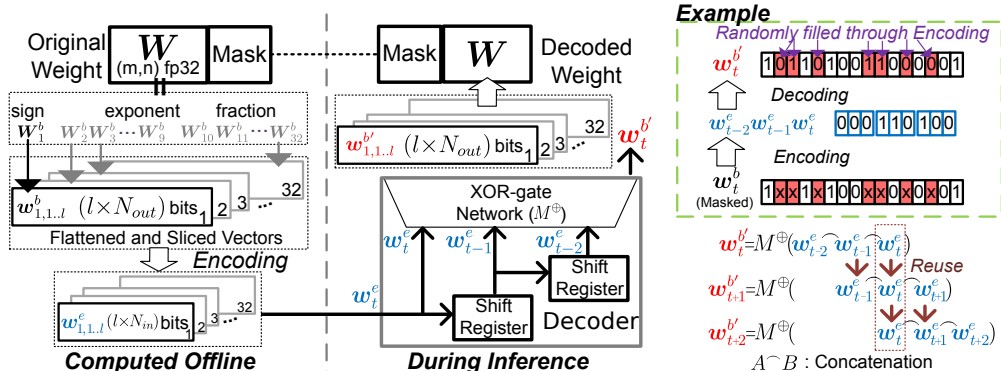

Figure 6: Proposed fixed-to-fixed sequential encoding/decoding scheme. Weight encoding is performed offline, and thus, complex computation for encoding is allowed. Encoded weights are decoded during inference through XOR-gate decoders that are best implemented by ASICs or FPGAs. Pruned weights are filled by random values during weight decoding.

**Decoding with an input sequence** For an XOR-gate decoder (as a non-sequential decoder) at time index $t$, an output vector $\boldsymbol{w}_t^{b'}$ ($\in \{0,1\}^{N_{out}}$) is a function of an input vector $\boldsymbol{w}_t^e$ ($\in \{0,1\}^{N_{in}}$) such that $\boldsymbol{w}_t^e$ is utilized only for one time index. In our proposed sequential encoding scheme, $\boldsymbol{w}_t^e$ is exploited for multiple time indices using shift registers. Specifically, we copy $\boldsymbol{w}_t^e$ to a shift register whose outputs are connected to the inputs of the next shift register. Let $N_s$ be the number of shift registers. In the proposed scheme, an XOR-gate decoder receives inputs from $N_s$ shift registers as well as $\boldsymbol{w}_t^e$. Then, as shown in Figure 6, a sequential decoder (consisting of an XOR-gate decoder and shift registers) produces $\boldsymbol{w}_t^{b'}$ as a function of an input sequence of ($\boldsymbol{w}_t^e$, $\boldsymbol{w}_{t-1}^e$, ...) while such a function can be described as $\boldsymbol{w}_t^{b'} = \boldsymbol{M}^\oplus(\boldsymbol{w}_t^e \frown \boldsymbol{w}_{t-1}^e \frown ... \frown \boldsymbol{w}_{t-N_s}^e)$ over $GF(2)$ where $A \frown B$ implies a concatenation of $A$ and $B$ and the sequence length is $(N_s + 1)$. Note that in Figure 6, an input vector $\boldsymbol{w}_t^e$ is reused $N_s$ times and an XOR-gate decoder accepts $(N_s + 1) \cdot N_{in}$ bits to yield $N_{out}$ bits (hence, $\boldsymbol{M}^\oplus \in \{0,1\}^{N_{out} \times ((N_s+1) \cdot N_{in})}$). Increasing $N_s$ enables 1) a larger XOR-gate decoder (without increasing $N_{in}$) that improves $E$ as demonstrated in Figure 4 and 2) multiple paths from an input vector to multiple output vectors (of various $n_u$) resulting in balanced encoding. Note that our XOR-gate decoder (or effectively memory decompressor) is best implemented by ASICs or FPGAs where each XOR requires only 6 transistors (Rabaey et al., 2004).

**Balanced encoding** Figure 7 illustrates a decoding process when $N_s$=1, $N_{in}$=3, and $N_{out}$=8. Each weight vector to be encoded contains a different number of unpruned weights. For $\boldsymbol{w}_t^{b'}$ at time index $t$ (in Figure 7), a less number of unpruned weights tends to enlarge search space for $\boldsymbol{w}_t^e$. On the other hand for $\boldsymbol{w}_{t+1}^{b'}$ at time index $(t + 1)$, a large number of unpruned weights would highly restrict search space for $\boldsymbol{w}_t^e$. Correspondingly, compared to the case when $\boldsymbol{w}_t^e$ is associated with only one $\boldsymbol{w}^{b'}$ block (i.e., non-sequential encoding with $N_s$=0), search space for $\boldsymbol{w}_t^e$ can be balanced as more $\boldsymbol{w}^{b'}$ blocks of various $n_u$ are correlated with $\boldsymbol{w}_t^e$. Note that as $\text{Var}[n_u]$ of one block increases, ac-

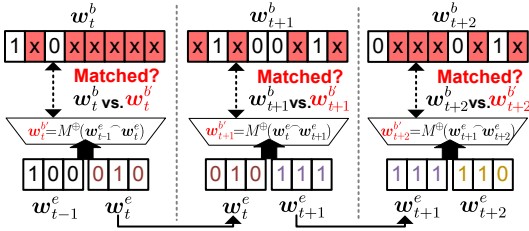

Figure 7: Sequential decoding example when $N_s$=1, $N_{in}$=3, and $N_{out}$=8. An input is utilized for $(N_s+1)$ time indices through shift registers.

cording to the central limit theorem, $N_s$ is required to be larger to maintain the balance of encoding capability of each $\boldsymbol{w}_t^e$. As we demonstrate in the next section, under the variation on $n_u$, even small non-zero $N_s$ can enhance $E$ substantially while increasing $N_{in}$ (and $N_{out}$ while $N_s$=0) improves $E$ only marginally (as described in Figure 4).

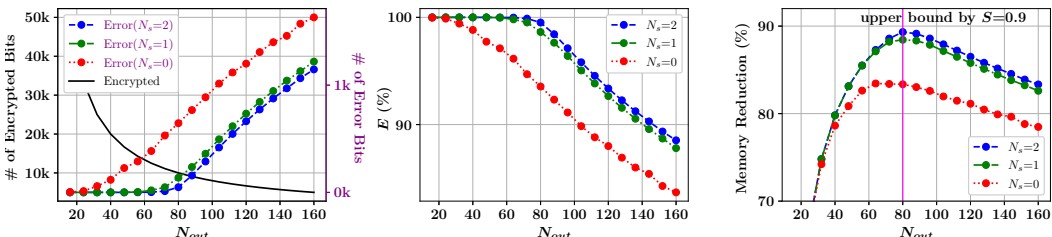

Figure 8: Impact of $N_s$ with various $N_{out}$ using 1M random bits, $N_{in} = 8$, and $S$=0.9.

**Encoding algorithm**   In the case of a non-sequential encoding scheme, because of one-to-one correspondence between $\boldsymbol{w}_t^e$ and $\boldsymbol{w}_t^{b'}$ through $\boldsymbol{M}^{\oplus}$, encoding can be performed independently for each block (e.g., for a given masked $\boldsymbol{w}_t^b$, we select one matching $\boldsymbol{w}_t^{b'}$ out of all available $\boldsymbol{w}_t^{b'}$ sets by a decoder). Such block-wise encoding, however, is not applicable to a sequential encoding scheme that needs to consider the whole sequence of $\boldsymbol{w}^b$ blocks to find an input sequence fed into a decoder. Suppose that we find a particular output sequence matching $l$ blocks after exploring all feasible output sequences provided by a generator and the input sequence $(\boldsymbol{w}_1^e, \boldsymbol{w}_2^e, ..., \boldsymbol{w}_l^e)$, the time-complexity of encoding would be $\mathcal{O}(2^{N_{in} \cdot l})$. Fortunately, sequential decoding operations shown in Figure 7 can be models as a hidden Markov model, where each state is represented by concatenating $\boldsymbol{w}_t^e$, $\boldsymbol{w}_{t-1}^e$, ..., $\boldsymbol{w}_{t-N_s}^e$ and there are $2^{N_{in}}$ paths for the next state transitions (Viterbi, 1998). Consequently, the time- and space-complexity can be reduced to be $\mathcal{O}(2^{N_{in} \cdot (N_s+1)} \cdot l)$ by dynamic programming that computes the maximum-likelihood sequence in a hidden Markov model (Forney, 1973). For details of our encoding algorithm, the reader is referred to Appendix E.

**Lossless compression**   Any random number generators cannot produce outputs perfectly matching all unpruned weights (i.e., $E$ is always less than 100%). To correct unmatched outputs of a random number generator (by flipping $0 \leftrightarrow 1$) in order to enable lossless compression, the locations of all unmatched weight bits need to be recorded. Note that if $E \approx 1$, the number of unmatched weight bits is a lot smaller than the number of encoded weight bits. If that is the case, such correction information can be stored in a separate on-chip memory that can be independently accessed without disturbing decoding operations. We propose a format representing such correction information in Appendix F where each unmatched weight bit requires $N_c$ bits in the correction format ($N_c$ is around 10 in Appendix E). Taking into account a compression ratio of a generator given as $N_{out}/N_{in}$ and additional correction information (using $N_c$ bits per one unmatched weight bit), a binary weight matrix $\boldsymbol{W}^b \in \{0,1\}^{m \times n}$ can be compressed into $(\frac{N_{in}}{N_{out}} mn + N_{err})$ bits when $N_{err} = N_c \times$(# of unmatched bits) $= N_c mn (1-S)(1-E)$. Subsequently, memory reduction can be expressed as

$$\begin{aligned} \text{Memory Save} &= 1 - ((N_{in}/N_{out})mn + N_{err})/(mn) \\ &= 1 - (1-S)(1 + (1-E)N_c), \end{aligned} \tag{2}$$

when $N_{in}/N_{out}$ is given as $(1-S)$. Thus, memory reduction approaches $S$ when $E$ approaches 1. For the overall design complexity analysis of the proposed compression, refer to Appendix G.

## 5   EXPERIMENTAL RESULTS

In this section, we demonstrate the encoding capability of our proposed sequential encoding techniques using synthetic random data and NNs pruned by various pruning methods. Even though various heuristic algorithms can be suggested, we adopt a simple dynamic programming technique for weight encoding that minimizes the number of unmatched weight bits. For our experiments, $N_{in}$ is selected to be 8 such that we feed a decoder on a byte-level.

### 5.1   SYNTHETIC RANDOM DATA ($\boldsymbol{N_{in}}$=8)

**Setup**   We generate a random weight 1-D vector of 1M bits. We also create a random masking data of 1M bits in which the percentage of zeros equals $S$. $\boldsymbol{M}^{\oplus}$ matrix (formulating the structure of

an XOR-gate decoder) basically needs to be designed to maximize the randomness among outputs. Measuring randomness, however, is challenging and such measurement may not be highly correlated to $E$. Alternatively, we try numerous random $\boldsymbol{M}^{\oplus}$ matrices and choose a particular $\boldsymbol{M}^{\oplus}$ of the highest $E$. Specifically, for a given set of $N_{in}$ and $N_{out}$, an element of $\boldsymbol{M}^{\oplus} \in \mathbb{R}^{N_{out} \times ((N_s+1) \cdot N_{in})}$ is randomly assigned to 0 or 1 with equal probability. Then, $E$ of those random $\boldsymbol{M}^{\oplus}$ matrices is estimated by using given random binary weight vectors and masking data. The best $\boldsymbol{M}^{\oplus}$ providing the highest $E$ (for a given set of $N_{in}$ and $N_{out}$) is then utilized for our experiments.

**Compression capability**  Figure 8 demonstrates the impact of $N_s$ on $E$ and corresponding memory reduction(%) with various $N_{out}$ and 1M random bits when $N_{in}$=8 and $S$=0.9. Regardless of $N_s$, as $N_{out}$ increases (i.e., the compression ratio (=$N_{out}/N_{in}$=$N_{out}/8$) of a decoder increases), the number of encoded bits is reduced while the number of unmatched (error) bits increases. Because of such a trade-off between the number of encoded bits and the number of error bits, there exists a certain $N_{out}$ that maximizes the memory reduction. Note that compared to a non-sequential encoding (of $N_s$=0), sequential encoding (even with small $N_s$) significantly reduces the number of error bits and maintains high $E$ (of almost 100%) until $N_{out}$ reaches $N_{in} \times (1/(1-S))$=80. Indeed, memory reduction becomes highest (=89.32%) when $N_s$ is highest and $N_{out}$ is 80 in Figure 8. As we discussed for lossless compression in Section 5, 89.32% of memory reduction is close to $S$(=90%) that is the maximum memory reduction obtained when $E \approx 1$. In the remainder of this paper, $N_{out}$ is given as $N_{in} \times (1/(1-S))$ to maximize the memory reduction.

**Impact of $S$ on memory reduction**

Table 1 presents memory reduction using various $S$. For a certain $S$ in Table 1, as $N_s$ increases, the difference between $S$ and memory reduction decreases. In other words, regardless of $S$, sequential encoding/decoding principles are crucial for memory reduction to approach $S$ which is the maximum. Increasing $S$ also facilitates memory reduction to approach

Table 1: Memory reduction (%) using 1M random bits and various $N_s$ and $S$ when $N_{in} = 8$ and $N_{out}$=$N_{in} \cdot 1/(1-S)$.

| $N_s$ \ $S$ | 60.0% | 70.0% | 80.0% | 90.0% |
|---|---|---|---|---|
| 0 | 38.6% | 53.8% | 67.9% | 83.5% |
| 1 | 55.9% | 67.4% | 77.5% | 88.5% |
| 2 | 58.4% | 69.1% | 78.9% | 89.3% |

$S$ in Table 1 as described in Eq. 2 where increasing $S$ with a constant $E$ enhances memory reduction.

**Inverting technique**  So far, we assumed that a binary weight matrix holds equal amounts of zeros and ones. A few representations such as binary-coding-based quantization (Rastegari et al., 2016; Xu et al., 2018) and signed INT8 (Jacob et al., 2018) would inherently justify such an assumption. However, there exist exceptional representations as well (e.g., FP32). We conduct experiments to find a relationship between the ratio of zeros and $E$ using a random weight vector as shown in Figure 9. $E$ increases if a substantial amount of zeros are employed as unpruned weight bits, because of a higher chance to find trivial inputs (of all zeros fed into XOR operations) to produce zero outputs. Hence, to improve $E$ (especially when $N_s$ is low), we propose an inverting technique where an entire binary weight vector is inverted if the ratio of zeros is less than 50%.

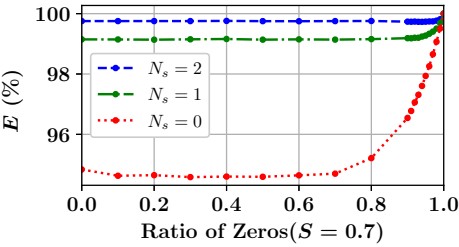

Figure 9: $E$ with various ratio of zero in a random vector.

## 5.2 SPARSE TRANSFORMER AND RESNET-50 ($N_{in}$=8)

We measure compression capability of our proposed sequential encoding scheme using sparse Transformer (Vaswani et al., 2017) on WMT'14 en-de dataset and ResNet-50 (He et al., 2016) on ImageNet. Those two models[1] in FP32 format are pruned by various methods including magnitude-based one (Han et al., 2015), L0 regularization (Louizos et al., 2018), and random pruning (Gale et al.,

---

[1]https://github.com/google-research/google-research/tree/master/state_of_sparsity

Table 2: $E$ and memory reduction of sparse Transformer and ResNet-50 pruned by two different pruning methods. When $N_s$ is 0 or 1, inverting can be applied to a layer if unpruned weights accommodate more zeros than ones. Inverting has no effect for weights of signed INT8.

| Model | $S$(Method) | $E$ (%) (Max: 100%) | | | Memory Reduction (%) (Max: $S$) | | |
|---|---|---|---|---|---|---|---|
| | | Non-Seq. $N_s$=0(Inv.) | Sequential $N_s$=1(Inv.) | $N_s$=2 | Non-Seq. $N_s$=0(Inv.) | Sequential $N_s$=1(Inv.) | $N_s$=2 |
| Transformer *WMT14 en-de* (FP32) | 70%(Mag.) | 93.8(94.5) | 98.0(98.3) | **98.7** | 50.3(52.4) | 63.1(63.8) | **65.3** |
| | 70%(Rand.) | 94.6(95.2) | 99.2(99.3) | **99.8** | 52.8(54.6) | 66.6(66.8) | **68.3** |
| | 90%(Mag.) | 92.6(93.9) | 97.6(97.9) | **98.4** | 82.4(83.7) | 87.4(87.7) | **88.2** |
| | 90%(Rand.) | 93.7(94.5) | 98.7(98.9) | **99.5** | 83.5(84.3) | 88.5(88.7) | **89.3** |
| ResNet-50 *ImageNet* (FP32) | 70%(Mag.) | 94.4(95.0) | 98.6(98.7) | **99.1** | 52.2(54.2) | 64.7(65.3) | **66.5** |
| | 70%(Rand.) | 94.6(95.1) | 99.1(99.2) | **99.7** | 52.7(54.2) | 66.5(66.7) | **68.3** |
| | 90%(Mag.) | 92.7(93.7) | 97.3(97.6) | **98.1** | 82.5(83.5) | 87.1(87.4) | **87.9** |
| | 90%(Rand.) | 92.7(93.5) | 97.6(97.9) | **98.7** | 82.5(83.3) | 87.5(87.7) | **88.6** |
| ResNet-50 *ImageNet* (Signed INT8) | 70%(Mag.) | 93.9(N/A) | 98.5(N/A) | **99.1** | 50.9(N/A) | 64.5(N/A) | **66.4** |
| | 70%(Rand.) | 96.2(N/A) | 99.7(N/A) | **99.9** | 57.6(N/A) | 68.3(N/A) | **69.0** |
| | 90%(Mag.) | 92.4(N/A) | 97.1(N/A) | **98.0** | 82.2(N/A) | 86.9(N/A) | **87.8** |
| | 90%(Rand.) | 93.5(N/A) | 98.2(N/A) | **99.2** | 83.3(N/A) | 88.0(N/A) | **89.0** |

Table 3: Coefficient of variation of $n_u$ and $E$ of two selected layers of the Transformer pruned by random, magnitude-based, or L0 regularization pruning method.

| Pruning Method | Target $S$ | Layer: dec3/self_att/q ($512 \times 512$), FP32 | | | | Layer: dec3/ffn2 ($2048 \times 512$), FP32 | | | |
|---|---|---|---|---|---|---|---|---|---|
| | | Coeff. of Var. ($n_u$) | $E$ (%) $N_s$=0 | $N_s$=1 | $N_s$=2 | Coeff. of Var. ($n_u$) | $E$ (%) $N_s$=0 | $N_s$=1 | $N_s$=2 |
| Random | | 0.299 | 94.6 | 99.2 | **99.8** | 0.303 | 94.6 | 99.2 | **99.8** |
| Mag. | 0.7 | 0.324 | 94.5 | 98.9 | **99.6** | 0.366 | 94.1 | 98.3 | **98.9** |
| L0 Reg. | | 0.347 | 94.5 | 99.0 | **99.6** | 0.331 | 94.3 | 98.7 | **99.2** |

2019) (also variational dropout (Molchanov et al., 2017) in Appendix H). For the ResNet-50 model (on ImageNet), we also consider signed INT8 format (Jacob et al., 2018). Table 2 presents $E$ and memory reduction when every layer of the Transformer and ResNet-50 is pruned by the same pruning rate $S$. Both $E$ and memory reduction are significantly improved by increasing $N_s$. Even compared to the case when inverting technique is applied to non-sequential encoding ($N_s$=0), we observe that sequential encoding ($N_s$>0) without inverting yields a lot higher compression capability. Note that the compression capabilities of random pruning and magnitude-based pruning methods are similar in Table 2 such that our experiments with synthetic random data are justified. Such justification is also verified in Table 3 that is achieved by using two selected layers of the Transformer. Compared to random pruning, magnitude-based and L0 regularization pruning methods exhibit somewhat lower $E$ that is related to higher coefficients of variation of $n_u$. See Appendix H for additional results.

All in all, our proposed encoding method designed in the context of random pruning is also effective for other fine-grained pruning methods. Through various cases including synthetic data and benchmark models, we demonstrated the superiority of the proposed encoding scheme to previous fixed-to-fixed compression methods including a non-sequential XOR-gate decoder of $N_s$=0 (Kwon et al., 2020) and a Viterbi-based encoder structure where $N_{in}$ is limited to be 1 (Ahn et al., 2019).

# 6 CONCLUSION

In this paper, we proposed a sequential encoding scheme that is a useful fixed-to-fixed compression for sparse NNs. We studied the maximum compression ratio using entropy based on the strategy of mapping a weight block into a small number of symbols. We also investigated random number generators as a practical fixed-to-fixed decoder using an XOR-gate decoder. Random number generators can improve compression capability if input vectors are reused for multiple time indices through shift registers.

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

## A    MEMORY BANDWIDTH WITH FIXED-TO-VARIABLE SPARSITY FORMAT

Memory bandwidth is expressed as the access rate (usually in units of bytes/second) at which the data can be read from or written into memory. While the number of memory transactions can be reduced a lot by pruning weights, the utilization of memory bandwidth is affected by the variability of the length of data to be accessed during one transaction (Yu et al., 2017). In the case of fixed-to-fixed sparsity representation, since all encoded blocks have the same size, full memory bandwidth is utilized. On the other hand, in the case of fixed-to-variable weight representation, encoded blocks have variable sizes such that memory bandwidth can be wasted. To be more specific, let $n_w$, $n_b$, and $S$ be the number of weights to be encoded, the number of unpruned weights in a block, and sparsity, respectively. Assuming that pruning a weight is a Bernoulli trial, for CSR format, we obtain

$$\mathbb{E}[n_b] = n_w \times (1 - S) \tag{3}$$
$$\mathrm{Var}[n_b] = n_w \times S \times (1 - S). \tag{4}$$

Thus, the coefficient of variation (or relative standard deviation) is given as

$$\frac{\sqrt{\mathrm{Var}[n_b]}}{\mathbb{E}[n_b]} = \frac{1}{\sqrt{n_w}}\sqrt{\frac{S}{1-S}} \tag{5}$$

, which increases as $S$ increases ($0 < S < 1$). In other words, for fixed-to-variable sparsity format, more memory bandwidth is wasted as more weights are pruned (as shown in Figure 1). Note that for fixed-to-fixed sparsity format, we have $\mathrm{Var}[n_b] = 0$, and thus, the memory bandwidth is not wasted at all.

## B    PERFORMANCE OF SPARSE MATRIX MULTIPLICATION USING CSR FORMAT

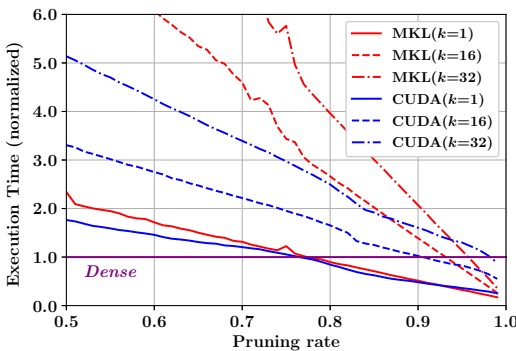

Figure S.10: Normalized execution time of multiplying a ($2048 \times 2048$) sparse matrix with a ($2048 \times k$) dense matrix.

For inference of NNs, it is known that performance (including latency and throughput) is dominated by matrix multiplications. Figure S.10 presents performance when a ($2048 \times 2048$) sparse matrix (of CSR format) is multiplied by a ($2048 \times k$) dense matrix when $k$ is usually small for inference with small batch size. MKL library (operated by i7-7700 @ 3.6GHz) and CUDA 10.2 library (performed by nVIDIA V100) perform sparse matrix multiplications whose execution times are normalized with respect to corresponding dense matrix multiplications (i.e., using a dense ($2048 \times 2048$) matrix). From Figure S.10, it should be noted that even with a high compression ratio (due to high pruning rate), if CSR format is adopted, sparse matrix multiplications can be even slower than dense matrix multiplication. Thus, proposing a regular format after fine-grained pruning is critical for parallel computing to achieve performance gain by pruning.

## C    RELATED WORKS

This section describes the previous works regarding how to represent sparsity after fine-grained weight pruning. As mentioned in Section 1, high sparsity and compression ratio can be expected

by fine-grained (unstructured) pruning, which prunes individual weights at randomly distributed locations.

To represent sparse weights with a reduced memory footprint, recording unpruned weights associated with (binary) masking information may be the simplest way. However, for such a case, a reconstruction process is required right before matrix multiplications during inference and it is challenging to be parallelizable due to its random locations (i.e., the number of pruned weights to be processed by a computation unit should highly vary as mentioned in Appendix A). As efforts to save memory footprints with fine-grained sparsity, by large, there have been two lines of researches, namely, fixed-to-variable and fixed-to-fixed formats. The Compressed Sparse Row (CSR) format is a well-known example of fixed-to-variable formats. Since DeepCompression (Han et al., 2016b) utilized the CSR format for representing sparse neural networks (NNs), most of the acceleration library kernels and computing systems have supported CSR data formats along with related APIs for CSR format (e.g. *cusparse* library in *CUDA*). Based on recording unpruned values and relational indices row-wise, the CSR format can lead to increased parallelism for computing sparse neural networks. However, because the number of unpruned weights in each row still varies, CSR inherently leads to irregular memory access patterns. SpMM or SpMV operations based on CSR format (even when including block CSR or advanced hardware design for CSR (Han et al., 2016a)) cannot still achieve full (or high) memory bandwidth utilization of computing systems, as numerous studies have raised related issues (Wen et al., 2016; Yu et al., 2017; Ahn et al., 2019; Zhou et al., 2018; Kwon et al., 2020; Gale et al., 2020) (we also reported similar concerns in Appendix B). Despite a highly reduced amount of parameters to be stored, pruned networks in an unstructural manner have not been fully employed by the currently available commercialized computing systems.

Fixed-to-fixed data compression, on the other hand, can achieve fully-parallelizable computations along with memory-saving formats and higher memory bandwidth because the length of compressed data is supposed to be ideally equal among any subsets of compressed data. By adopting compression approaches that have been widely used in well-established engineering areas (e.g., digital communication, VLSI testing, and so on), Viterbi-based compression (Ahn et al., 2019) and XOR-gates-based compression (Kwon et al., 2020) have been proposed. Note that despite regular memory access patterns, compression methods of Ahn et al. (2019) require a heavy re-training process to acquire proper weight bits and masks while presenting the limitation that the compression ratio should be fixed to be integer values. As for the work of (Kwon et al., 2020), the encoding efficiency of previous XOR-gates-based compression is significantly degraded due to combinational encoding algorithm and inefficiency of patch systems.

Our proposed work can be regarded as an extended study of XOR-gates-based compression: 1) this paper verifies that typical data representations (e.g., FP32 and INT8) for DNNs can also be compressed by the XOR-gate decoder (additionally, the inverting technique can boost the encoding efficiency), 2) By using the proposed sequential encoding/decoding schemes, encoding efficiency $E$ can closely approach the theoretical upper bound of fixed-to-fixed sparsity formats, and 3) Our encoding algorithm can explore encoded bits according to sequential decoding within the limited size of XOR-gate decoder while Kwon et al. (2020) proposed a simple heuristic algorithm assuming combinational decoding only (i.e., $N_s = 0$).

## D    FUNDAMENTAL LIMITS OF COMPRESSION

We are interested in the upper bound of compression ratio that can be analyzed by entropy (while allowing fixed-to-variable compression). Then, when we suggest a fixed-to-fixed compression scheme, we can estimate how close the compression capability of a fixed-to-fixed scheme is to the maximum.

Entropy presents the minimum average number of bits to represent an event when a probability distribution of those events is provided (Morelos-Zaragoza, 2006). To investigate the entropy of pruned weight blocks (and the maximum compression ratio correspondingly), a block of bits is assigned to a symbol such that a probability distribution of symbols minimizes the entropy. In other words, symbol assignment is designed to minimize the average number of bits (i.e., entropy to represent symbols) that is given as

$$H = -\sum_{i=1}^{n} p_i \cdot \log_2 p_i, \tag{6}$$

where $n$ is the total number of symbols and $p_i$ is the occurrence probability of each symbol.

**Symbol assignment** We concatenate all of $k$-th bits of weights into a group and produce $n_w$ groups when $n_w$ is the number of bits to represent a weight (e.g., $n_w$ is 32 for single-precision and 8 for INT8) and $1 \leq k \leq n_w$. Symbol assignment is performed in each group independently without referring to other groups. Suppose that a certain block from one of the groups is given as $\{0\times\times1\}$ and corresponding mask bits (that are shared by all $n_w$ groups) are $\{1001\}$ (0 means masking). By filling up $\times$ with 0 or 1, $\{0\times\times1\}$ is selected to be assigned to one of 4 symbols (i.e., $\{0001\}$, $\{0011\}$, $\{0101\}$ or $\{0111\}$) while such a symbol selection decides entropy. In the following two examples (where $n_b$=4), we illustrate the symbol assignment method that minimizes the entropy.

**Entropy ($n_u$ = 1, e.g., $\{\times\times\times0\}$, $\{\times1\times\times\}$)** In this case, every block (of $n_b$=4 and $n_u$=1) can be assigned to either one of two symbols ($\{0000\}$, $\{1111\}$). Since $P(0000) = P(1111) = 0.5$, from Eq. 6, $H = -(0.5 \times (-1) + 0.5 \times (-1)) = 1$. Thus, a block of $n_b$=4 and $n_u$=1 can be compressed into 1 bit that indicates one of two symbols. There are many other sets of two symbols to meet $H = 1$, such as ($\{0010\}$, $\{1101\}$) and ($\{1010\}$, $\{0101\}$).

**Entropy ($n_u$ = 2, e.g., $\{\times01\times\}$, $\{10\times\times\}$)** A set of symbols should meet the following requirement: if we choose random $n1$ and random $n2$ ($1 \leq n1, n2 \leq n_b, n1 \neq n2$) and collect $\{n1$-th bit, $n2$-th bit$\}$ of each symbol, then, each of $\{00, 01, 10, 11\}$ should appear in the collection. Under such a constraint, after a careful investigation using a full search, the minimum number of symbols is 5 to represent all blocks of $n_b$=4 and $n_u$=2. An example set of 5 symbols with corresponding occurrence probability to minimize entropy is as follows: $P(0000) = 6/24$, $P(1110) = 6/24$, $P(0101) = 5/24$, $P(1001) = 4/24$, and $P(0011) = 3/24$. Then, from Eq. 6, $H$ is approximately 2.28 (bits) attainable through fixed-to-variable compression. On the other hand, for a fixed-to-fixed scheme, a block (of $n_b$=4 and $n_u$=2) is compressed into 3 bits to represent one of 5 symbols.

For $n_u$=3, the minimum number of symbols is 8 and a block can be compressed into 3 bits. All in all, **when $n_u$ is fixed across blocks, $H$ can be equal to or slightly higher than $n_u$.**

# E    ENCODING ALGORITHM

In this section, we describe our encoding algorithm based on dynamic programming algorithm that can minimize the number of unmatched bits. Our encoding algorithm presents time-complexity as $\mathcal{O}(l \cdot 2^{N_{in} \cdot (N_s+1)})$ and space-complexity as $\mathcal{O}(2^{N_{in} \cdot N_s})$. Thus, even though increasing $N_{in}$ and $N_s$ enhances $E$, $(N_{in} \times N_s)$ is empirically limited to be less than 26 under the constraint of 32GB memory of a single GPU. For each binary weight $\boldsymbol{W}_i^b (0 \leq i \leq n_w)$, the function ENCODING generates the encoded vectors $w_{i,(1..l+N_s)}$. XOR-gate decoder ($\boldsymbol{M}^\oplus$) is pre-determined and fixed for inference. Note that the number of encoded vectors ($\boldsymbol{w}_{1..(l+N_s)}^e$) is $l + N_s$, not $l$. $w_1^e$ and $w_2^e$ are pre-determined as $BIN(0)$ and used for encoding the first binary weight vector.

---

**Algorithm 3:** Encoding algorithm when $N_s = 2$. For a binary matrix $\boldsymbol{W}_i^b$, the ENCODING function generates encoded bit vectors. For varied $N_s$, the number of for-loop statements for $i^t$ (e.g. line 34-36) and the dimensions of arrays ($dp$ and $path$) are changed to $N_s + 1$.

---

**Parameters** $\boldsymbol{W}_i^b$ is a binary weight vector to be encoded. $MASK$ is pruning mask information for $\boldsymbol{W}_i^b$. $dp$ is $(N_s+1)$-dimensional array. $dp[t][a][b]$ stores the minimum number of error bits when BIN(a) and BIN(b) are fixed as $w_t^e$ and $w_{t-1}^e$. $path$ is $(N_s+1)$-dimensional array for history.

**Additional Functions** SIZE($\boldsymbol{W}$) returns the number of parameters in $\boldsymbol{W}$. RESHAPE($ar$, $shape$) returns same data ($ar$) with specified shape, $shape$. INIT($ar$, $init$) sets a value $init$ to all elements in $ar$. BIN($dec$) returns a binary value of $dec$. DEC($bin$) returns a decimal value of $bin$.

**Function** ERR_NUM($x, y, mask$):
    $nErr \leftarrow 0$
    **for** $i \leftarrow 0$ **to** $N_{out} - 1$ **do**
        **if** $mask[i] \neq 0$ *and* $x[i] \neq y[i]$ **then**
            $nErr \leftarrow nErr + 1$
    **return** $nErr$

**Function** ENCODING($\boldsymbol{W}_i^b, MASK$):
    $l \leftarrow$ SIZE($\boldsymbol{W}_i^b$)$/N_{out}$
    $data \leftarrow$ RESHAPE($\boldsymbol{W}_i^b, [l, N_{out}]$) ▷ Slicing
    $mask \leftarrow$ RESHAPE($MASK, [l, N_{out}]$) ▷ Slicing
    INIT ($dp, INF$) ▷ Initialize for starting point
    $dp[N_s][0][0] \leftarrow 0$ and $\boldsymbol{w}_1^e, \boldsymbol{w}_2^e \leftarrow$ BIN(0), BIN(0)

    ▷ Find the minimum number of errors
    **for** $t \leftarrow N_s + 1$ **to** $l + N_s$ **do**
        **for** $i^t \leftarrow 0$ **to** $2^{N_{in}} - 1$ **do**
            **for** $i^{t-1} \leftarrow 0$ **to** $2^{N_{in}} - 1$ **do**
                **for** $i^{t-2} \leftarrow 0$ **to** $2^{N_{in}} - 1$ **do**
                    $out \leftarrow M^{\oplus}($BIN$(i^{t-2})^{\frown}$BIN$(i^{t-1})^{\frown}$BIN$(i^t))$
                    $n_{err} \leftarrow$ ERR_NUM($out, data[t], mask[t]$)
                    **if** $dp[t][i^t][i^{t-1}] > n_{err} + dp[t-1][i^{t-1}][i^{t-2}]$ **then**
                        $dp[t][i^t][i^{t-1}] \leftarrow n_{err} + dp[t-1][i^{t-1}][i^{t-2}]$
                        $path[t][i^t][i^{t-1}] \leftarrow$ BIN$(i^{t-2})$

    ▷ Find the last two(=$N_s$) encoded vectors
    $min_{err} \leftarrow INF$.
    **for** $i^t = 0$ **to** $2^{N_{in}} - 1$ **do**
        **for** $i^{t-1} = 0$ **to** $2^{N_{in}} - 1$ **do**
            **if** $min_{err} > dp[l + N_s][i^t][i^{t-1}]$ **then**
                $min_{err} \leftarrow dp[l + N_s][i^t][i^{t-1}]$
                $\boldsymbol{w}_{l+2}^e, \boldsymbol{w}_{l+1}^e \leftarrow$ BIN$(i^{t-1})$, BIN$(i^t)$

    ▷ Get encoded bits by following history array
    **for** $t \leftarrow l$ **to** $2N_s + 1$ **by** $-1$ **do**
        $\boldsymbol{w}_t^e \leftarrow path[t + N_s][$DEC$(\boldsymbol{w}_{t+2}^e)][$DEC$(\boldsymbol{w}_{t+1}^e)]$
    **return** $\{\boldsymbol{w}_1^e, \boldsymbol{w}_2^e, ..., \boldsymbol{w}_{l+N_s}^e\}$

---

# F MEMORY REDUCTION WITH LOSSLESS COMPRESSION

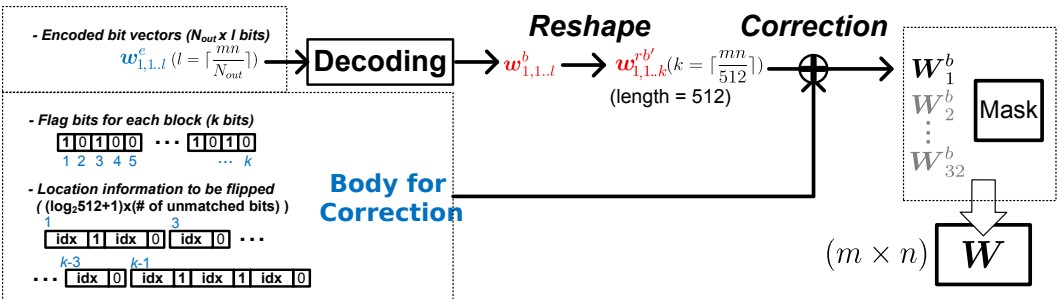

Figure S.11: Correction process for lossless compression. After the encoded (compressed) bit vectors are decoded, unmatched bits (that encoding could not target successfully) are flipped by correction information that records the locations of unmatched weight bits.

For lossless compression, the unmatched bits (error bits) should be corrected right after decoding procedures. Since the random number generator produces 0 or 1, the unmatched bits can be simply corrected by flipping.

To compute the memory reduction, we suggest block-wise correction logic to be conducted by flipping error bits in a $p$-length vector while each error location is given as a bit position inside a vector. As depicted in Figure S.11, when the block-wise correction is performed while a block size has $p$-length, the decoded vectors $\boldsymbol{w}_{1..l}^{b'}$ are reshaped to $\boldsymbol{w}_{1..k}^{rb'}(k = \lceil \frac{mn}{p} \rceil)$ and each reshaped vector $\boldsymbol{w}^{rb'}$ is corrected by corresponding error bit locations indicating which bit is to be flipped inside a block (of $p$-length).

The amount of compressed bits can be computed as

$$N_{in} \cdot \lceil \frac{mn}{N_{out}} \rceil + \lceil \frac{mn}{p} \rceil + (\log_2 p + 1) \times (\texttt{\# of unmatched bits}). \tag{7}$$

The first term is the number of bits of $\boldsymbol{w}_{1..l}^e$ as the compression results by sequential encoding. The second term is the number of flag bits while each flag bit indicates whether a non-zero number of unmatched bits exists in each $p$-length vector (as $E$ approaches 1, there are many $p$-length vectors that skip the correction step)). The third term includes locations of unmatched bits (inside a $p$-length block) to be flipped and an additional one bit to specify the end of the streaming error bit locations (i.e., '1' means the following $(\log_2 p)$ bits contains the next correction information of the same block). Thus, each block of $p$-length involves $(\log_2 512 + 1) \times (\text{\# of unmatched bits})$ for error bit locations.

## G    DESIGN COMPLEXITY OF THE PROPOSED COMPRESSION METHOD

**Encoding Process**    The time and space complexity of the encoding process (presented in Appendix E with corresponding algorithm description) is independent of inference performance since encoding is performed offline (thus, GPUs or CPUs would be fine to run the encoding algorithm). Encoding algorithm based on a dynamic programming technique as shown in Appendix E has the following space and time complexity.

- Time complexity: $\mathcal{O}(l \cdot 2^{N_{in} \cdot (N_s + 1)})$
- Space complexity: $\mathcal{O}(2^{N_{in} \cdot N_s})$

Algorithm 3 shown in Appendix E explores all possible $2^{((l + N_s) \cdot N_{in})}$ outputs of an XOR-gate decoder to produce an input vector that can minimize the number of unmatched bits. Note that a partial string of input (of an XOR-gate network) having $2^{(N_{in} \cdot (N_s + 1))}$ bits share $2^{N_{in} \cdot N_s}$ bits continuously through shift registers. As such, for the time index $t + 1$, search space (having the size of $2^{((t + 1 + N_s) \cdot N_{in})}$) is overlapped with the search space of the time index $t$ (having the size of $2^{((t + N_s) \cdot N_{in})}$) as much as $2^{t \cdot N_{in}}$. Accordingly, the time complexity at the time index $t + 1$ (that optimize the input) is reduced from $\mathcal{O}(2^{((t + 1 + N_s) \cdot N_{in})})$ to $\mathcal{O}(2^{((t + 1 + N_s) \cdot N_{in})}/2^{(t \cdot N_{in})}) = \mathcal{O}(2^{((1 + N_s) \cdot N_{in})})$. Then, since we iterate such operation $l$ times, the overall time complexity becomes $\mathcal{O}(l \cdot 2^{((1 + N_s) \cdot N_{in})})$. To exploit sharing computations between different time indices, we need to store intermediate results having the size of $2^{N_s \times N_{in}}$ which becomes the space complexity of Algorithm 3.

**Decoding Process**    As for decoding operations, we suggest that the decoding algorithm is best supported by a hardware design consisting of XOR gates (and a few shift registers). Hence, in this section, let us specifically argue hardware design issues of XOR-gate decoders.

- The strongest benefit of employing digital circuits (in the form of ASICs or FPGAs) to implement XOR gates is that all XOR gates can be performed simultaneously (unlike GPUs or CPUs where each core needs to simulate only a few XOR gates). Thus, all XOR operations of our proposed decoder are completed within just one clock cycle.
- $N_s$ (with shift registers) would increase the latency. Throughput, however, maintains to be the same regardless of $N_s$ through pipelining technique which is a basic hardware design principle.
- Overall, the design complexity (in terms of area overhead and latency) of XOR-gate decoders is extremely low (note that one XOR gate consumes only 6 transistors).
- XOR-gate decoders would work as memory decompressors (located in-between memory and computation logic). In the view of computational units that receive outputs of an XOR-gate decoder, then, the amount of memory is simply reduced while regular memory access patterns are not disturbed.
- Given $N_s$, an XOR-gate decoder requires $N_s$ additional clock cycles for the latency.
- Since $\boldsymbol{M}$ matrix has the size of $N_{out} \times N_{in}$ and an element of $\boldsymbol{M}$ is randomly filled with 0 or 1, the number of XOR gates is $(N_{out} \cdot N_{in}/2)$. Thus, the total number of transistors to design an XOR-gate decoder is $(3 \cdot N_{out} \cdot N_{in})$.
- Overall, we can provide full memory bandwidth (based on regular memory access patterns through fixed-to-fixed sparsity formats) while the overall hardware design cost is only marginal.
- While designing DNN inference accelerators is gaining increasing attention, our work can provide a new research direction.

# H  Additional Analysis on Experiments

In this section, we provide additional analysis and results for experiments in Section 6.2.

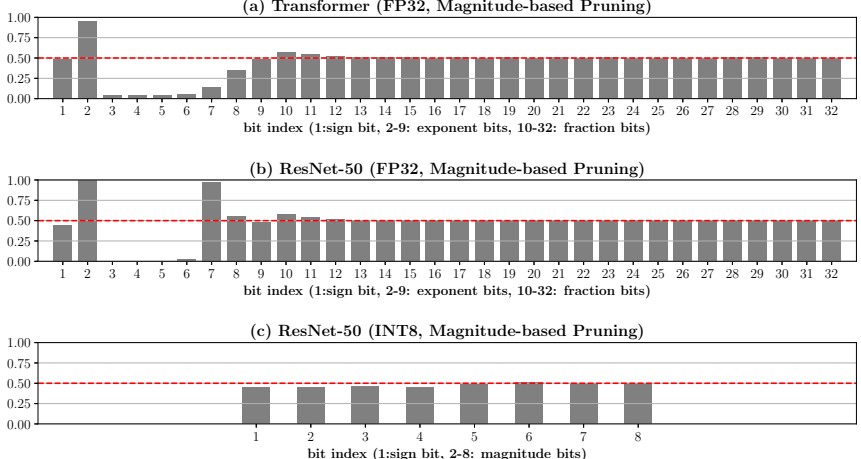

Figure S.12: Ratio of zeros when weights are divided into $k$ groups when $k$ is the bit-index (i.e., for FP32 number format, $k=1$ means a sign bit and $k=32$ means the least significant bit in mantissa). For models, Transformer(FP32), ResNet-50(FP32), and ResNet-50(INT8) are investigated. Most weights consist of similar amounts of 0s and 1s. For the exponent bits of FP32 models, there are noticeable skewed ratios of zeros (e.g. while most of the 2nd bits in Transformer are zero, most of 3rd, 4th, and 5th bits are ones.) because the range of exponent bits is limited according to characteristics of DNN models (e.g. regularization such as weight decay). We can gain additional efficiency by adopting the inverting technique for the FP32 models.

Table S.4: Coefficient of variation of $n_u$ and $E$ of selected layers of the Transformer pruned by random, magnitude-based, L0 regularization, or variational dropout pruning method.

| $N_{in}, N_{out}$ | SHAPE | $S$ | LAYER NAME | PRUNING METHOD | COEFF | $E$ (%) $N_s{=}0$ | $N_s{=}1$ | $N_s{=}2$ |
|---|---|---|---|---|---|---|---|---|
| | (512, 512) | 0.700 | DEC3/SELF_ATT/Q | RANDOM | 0.341 | 93.7% | 98.7% | 99.5% |
| | (2048, 512) | 0.700 | DEC3/FFN2 | RANDOM | 0.343 | 93.7% | 98.7% | 99.5% |
| | (512, 512) | 0.700 | DEC3/SELF_ATT/Q | MAGNITUDE | 0.390 | 93.4% | 98.2% | 99.0% |
| | (2048, 512) | 0.700 | DEC3/FFN2 | MAGNITUDE | 0.363 | 93.6% | 98.5% | 99.2% |
| (8, 26) | (512, 512) | 0.699 | DEC3/SELF_ATT/Q | L0 REG. | 0.476 | 92.7% | 97.6% | 98.7% |
| | (512, 512) | 0.698 | DEC3/FFN2 | L0 REG. | 0.467 | 92.7% | 97.7% | 98.8% |
| | (512, 512) | 0.705 | DEC5/SELF_ATT/K | VAR. DROPOUT | 0.303 | 94.5% | 99.1% | 99.7% |
| | (2048, 512) | 0.697 | DEC1/FFN1 | VAR. DROPOUT | 0.309 | 94.5% | 99.1% | 99.7% |
| | (512, 512) | 0.900 | ENC2/SELF_ATT/OUTPUT | RANDOM | 0.349 | 94.4% | 98.6% | 99.3% |
| | (2048, 512) | 0.900 | DEC5/FFN2 | RANDOM | 0.315 | 94.6% | 98.9% | 99.5% |
| | (512, 512) | 0.900 | ENC2/SELF_ATT/OUTPUT | MAGNITUDE | 0.516 | 92.5% | 97.0% | 98.0% |
| | (2048, 512) | 0.900 | DEC5/FFN2 | MAGNITUDE | 0.363 | 93.7% | 98.5% | 99.1% |
| (8, 80) | (512, 512) | 0.904 | ENC2/SELF_ATT/OUTPUT | L0 REG. | 0.331 | 94.3% | 98.7% | 99.2% |
| | (512, 512) | 0.896 | DEC5/FFN2 | L0 REG. | 0.347 | 94.5% | 99.0% | 99.6% |
| | (512, 512) | 0.906 | DEC2/SELF_ATT/V | VAR. DROPOUT | 0.770 | 89.4% | 93.6% | 94.3% |
| | (2048, 512) | 0.904 | DEC4/FFN1 | VAR. DROPOUT | 0.499 | 91.8% | 96.4% | 97.4% |

Table S.5: Coefficient of variation of $n_u$ and $E$ of selected layers of the ResNet-50 pruned by random, magnitude-based, or variational dropout pruning method.

| $N_{in}, N_{out}$ | SHAPE | $S$ | LAYER NAME | PRUNING METHOD | COEFF | $E$ (%) | | |
|---|---|---|---|---|---|---|---|---|
| | | | | | | $N_s=0$ | $N_s=1$ | $N_s=2$ |
| (8, 26) | (1,1,1024,256) | 0.700 | GROUP2_LAYER3_BN2 | RANDOM | 0.347 | 94.2% | 98.6% | 99.2% |
| | (1,1,256,1024) | 0.700 | GROUP3_LAYER5_BN3 | RANDOM | 0.334 | 94.3% | 98.8% | 99.5% |
| | (1,1,1024,256) | 0.700 | GROUP2_LAYER3_BN2 | MAGNITUDE | 0.505 | 93.2% | 98.0% | 98.9% |
| | (1,1,256,1024) | 0.700 | GROUP3_LAYER5_BN3 | MAGNITUDE | 0.428 | 93.8% | 98.4% | 99.0% |
| | (3,3,512,512) | 0.709 | GROUP2_LAYER3_BN2 | VAR. DROPOUT | 0.403 | 94.2% | 98.4% | 99.1% |
| | (1,1,256,1024) | 0.706 | GROUP3_LAYER5_BN3 | VAR. DROPOUT | 0.361 | 94.4% | 98.6% | 99.1% |
| (8, 80) | (3, 3, 256, 256) | 0.900 | GROUP3_LAYER3_BN2 | RANDOM | 0.683 | 92.3% | 96.8% | 98.0% |
| | (1, 1, 512, 2048) | 0.900 | GROUP4_LAYER0_BN3 | RANDOM | 0.407 | 92.9% | 97.4% | 98.1% |
| | (3, 3, 256, 256) | 0.900 | GROUP3_LAYER3_BN2 | MAGNITUDE | 0.303 | 94.6% | 99.2% | 99.8% |
| | (1, 1, 512, 2048) | 0.900 | GROUP4_LAYER0_BN3 | MAGNITUDE | 0.299 | 94.6% | 99.2% | 99.8% |
| | (3, 3, 256, 256) | 0.913 | GROUP3_LAYER3_BN2 | VAR. DROPOUT | 0.366 | 94.1% | 98.3% | 98.9% |
| | (1, 1, 512, 2048) | 0.896 | GROUP4_LAYER0_BN3 | VAR. DROPOUT | 0.324 | 94.5% | 98.9% | 99.6% |

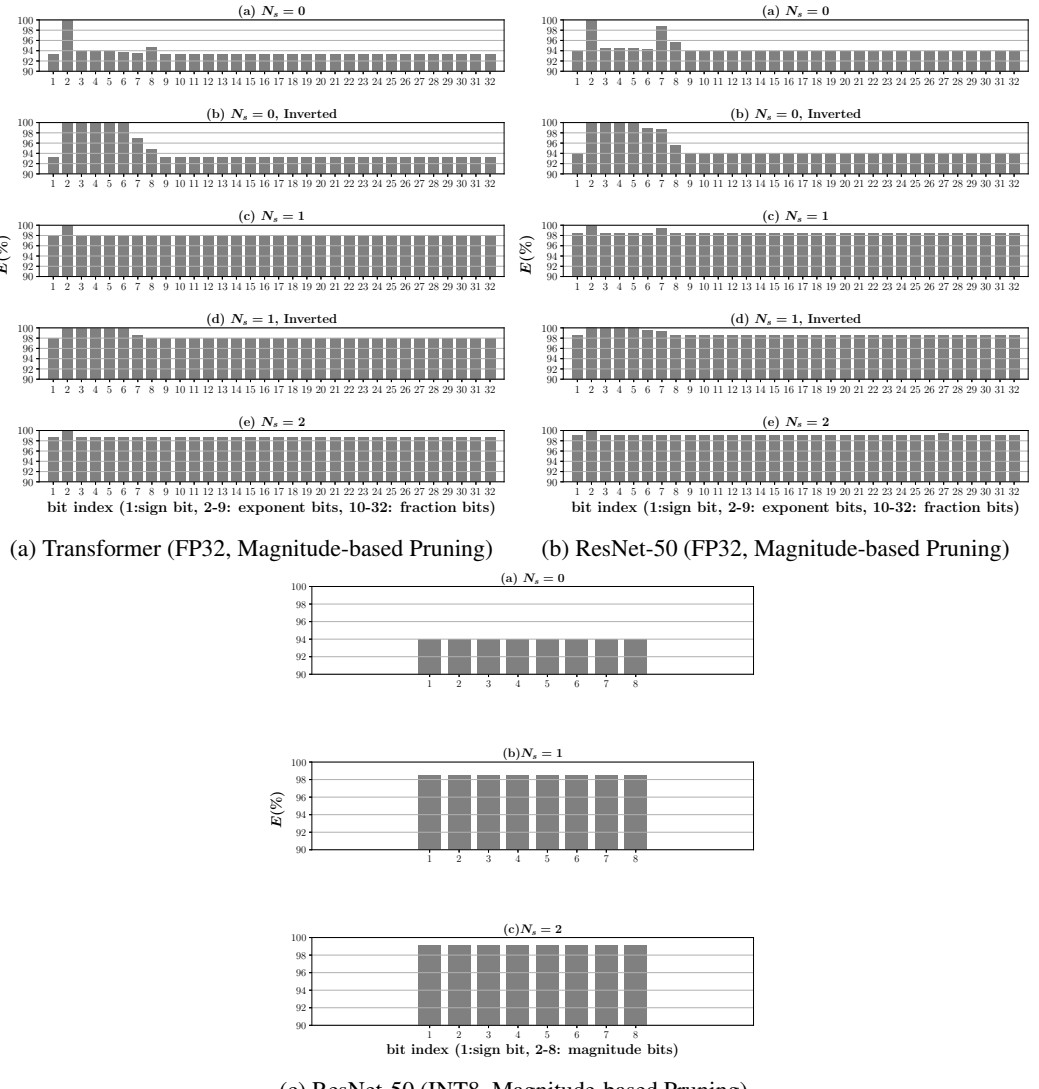

Figure S.13: $E$ of the Transformer and ResNet-50 (pruned by $S$=70%) measured for each bit index ($\leq$ 32 for FP32) individually with various $N_s$ while inverting technique is also considered . It can be observed that the inverting technique improves $E$ for $N_s = 0$ and $N_s = 1$. When $N_s = 2$, the improvement on $E$ is not noticeable.

