# OpenReview forum: "Encoding Weights of Irregular Sparsity for Fixed-to-Fixed Model Compression"
_ICLR.cc/2022/Conference — ICLR 2022 Poster_

### Official Review · Reviewer_Qcma · 2021-11-02

**Correctness:** 3
**Technical Novelty And Significance:** 3
**Empirical Novelty And Significance:** 3
**Recommendation:** 6
**Confidence:** 4

**Main Review:**

This work proposes a sequential fixed-to-fixed encoding scheme for sparse neural network weight encoding/decoding. The core component of the algorithm is shifted registers that expand the decoding window and extra code for recording unmatched bits.

Pros: The proposed shift register structure offers a much better improvement on encoding efficiency than increasing the coding block size. To my knowledge this algorithm is novel.

Cons:
It is unclear how much actual benefit this algorithm can offer as to my understanding this algorithm requires very high decoding computing and space complexity. With such kind of complexity CSR probably can also overcome its shortcoming. And such kinds of decoders would probably not fit in the very limited space of computing cores such as a GPU.

The paper is difficult for a reader that is not familiar with prior works such as Kwon et al., 2020 and Ahn et al., 2019. I only managed to comprehend this work after going through those two prior works. I suggest references to those works should be introduced in Sections 1 and 2 along with some related background info to help readers.

I suggest renaming the XOR-gate network to something like XOR-gate decoder, given this community generally associates “network” with neural networks.

P2: Please explain how you arrived at Figure 1A.

P2: Please explain “x with Mask” in Algorithm 2.

P2: Please precisely define memory bandwidth. What you described here seems vague.

P2: “Since higher sparsity leads to higher variance on pruned weights in a block”. Please clarify this sentence. Are you sure this is correct?

P3, last line: “(masked) weights”. Do you mean weights not subjected to pruning? Shouldn’t such weights be called not masked weights?

Figure 4: “S is entire pruning rate”, I believe “entire” here is not necessary and is causing confusion.

P4: A reference to GF(2) is probably necessary.

P4; the sentence just above Section 3.1. : I believe the logic of this sentence is not valid. Even though the conclusion is correct.

P7: “(Nin × Ns) is empirically limited to be less than 26” What you mean by this? If you chose to have $N_in=8$, does that leave you with $N_s<3$? If you need 32GB memory for decoding, is this algorithm applicable in real-world applications?

Table2 and 3 probably contain too much redundant information. I suggest removing some results so there will be enough space for improved clarity.

**Summary Of The Paper:**

This work proposes a sequential fixed-to-fixed encoding scheme for sparse neural network weight encoding/decoding. The core component of the algorithm is shifted registers that expand the decoding window and extra code for recording unmatched bits.

**Summary Of The Review:**

Given the issues I raised in the main review I will not recommend this paper for publication for now. But I would be happy to change my evaluation if the authors can improve the clarity of the paper and address my concerns.

After rebuttal:
I raised my evaluation from 5 to 6, given the improvement included in the new revision. However, there are still two main issues I want to point out. First, the authors built their work upon Kwon20 and Ahn19, but they only cite those two papers in the latter part of their manuscript.  And related works section is only included as an appendix section. This layout is unusual, and it made their manuscript hard to comprehend for an audience that is not familiar with the specific topic. Second, despite my repeated request for clarification on the computing complexity, the authors still only gave a vague description of the issue. I hope the authors can fix those two issues in their camera-ready if the paper is accepted.

---

> ### Author Response · Authors · 2021-11-17
> **Response to Reviewer Qcma**
>
> We appreciate your careful review.
>
> **R1: It is unclear how much actual benefit this algorithm can offer** \
> A1: Please refer to our responses to the other reviewers above. We agree with you that estimating the actual benefits of our proposed method is important. Full demonstration based on fixed-to-fixed sparsity formats, however, would require a few separate critical steps (namely, deriving theoretically feasible compression techniques, and then, designing corresponding hardware in the form of ASICs or FPGAs, not CPUs or GPUs). As the initial milestone of such a long journey, in our work, we demonstrated that numerous existing pruning methods can be efficiently represented by fixed-to-fixed sparsity formats that are inspired by theories of digital communication (specifically error correction coding). Our compression technique indeed demands a complex encoding process. But such a complex encoding process is performed offline such that inference performance has nothing to do with the computational complexity of encoding. Then, the decoding process can be efficiently operated (w/ small area overhead and high throughput) by simple XOR gates that have been already widely adopted by digital communication circuit design and VLSI testing. One XOR gate requires 6 transistors only while all XOR gates and shift registers can be run concurrently in ASICs or FPGAs. Since only weight decoding is related to inference performance, XOR-gate decoders would generate decoded weights with high performance (as demonstrated in the commercialized chips for digital signal processing). We acknowledge that simulating XOR gates in GPUs may not be the best way to fulfill our design requirements for weight decoding (since GPUs do not inherently accommodate such XOR-gate decoders).
>
> **R2: I suggest references in Sections 1 and 2** \
> A2: We added Appendix C to introduce related works.
>
> **R3: I suggest renaming the XOR-gate network.** \
> A3: Thanks a lot for your suggestion. Throughout the manuscript, we replaced “XOR-gate network” with “XOR-gate decoder” as you suggested.
>
> **[Additional comments]** \
> **To address your concerns of (Q1), (Q3), and (Q5), we added Appendix A to clarify potentially confusing discussions on memory bandwidth.
>
> **(Q1) How you arrived at Figure 1A?** \
> (A1)  Observations and empirical data similar to Figure 1A have been reported by numerous publications (i.e., SpMM or SpMV based on CSR format cannot achieve full memory bandwidth utilization) as discussed in Appendix C.
>
> **(Q2) Explain “x with Mask” in Algorithm 2.** \
> (A2) For Algorithm 2, we assume that the weight W does not involve “don’t care” values that need to be masked. In other words, pruned weights need to be replaced with zeroes where mask information is necessary.
>
> **(Q3) Define memory bandwidth.** \
> (A3) Memory bandwidth is expressed as the access rate (usually in units of bytes/second) at which the data can be read from or written into memory. For more details, please refer to Appendix A.
>
> **(Q4) “Since higher sparsity leads to ....”. Is this correct?** \
> (A4) In the case of fixed-to-fixed sparsity representation, since all encoded blocks have the same size, full memory bandwidth is utilized. On the other hand, in the case of fixed-to-variable weight representation, encoded blocks have variable sizes such that memory bandwidth can be wasted. The coefficient of variation (or relative standard deviation) on the number of unpruned weights increases as pruning rate increase, as discussed in Appendix A.
>
> **(Q5) P3, (masked) weights: Do you mean weights not subjected to pruning?** \
> (A5) Following the fine-grained pruning method, for a weight block, some weights are to be pruned and are to be eventually masked. Accordingly, we revised the corresponding sentence ((masked) is corrected to be (partially masked)).
>
> **(Q6) Figure 4: “entire” here is not necessary** \
> (A6) We agree. We removed the word “entire” for Figure 4.
>
> **(Q7) A reference to GF(2)** \
> (A7) Thanks for your suggestion. We added a relevant reference right after “GF(2).”
>
> **(Q8) P4; the sentence just above Section 3.1.: logic is not valid.** \
> (A8) We agree. We removed that corresponding sentence and left only essential information.
>
> **(Q9) P7; (Nin x Ns) is empirically limited to be less than 26. What do you mean by this?** \
> (A9) To avoid any possible confusion, we left fundamental arguments only in P7 and moved the corresponding sentence to Appendix E with additional information to explain how such particular constraints have been derived in our experimental settings.
>
> **(Q10) Table 2 and 3 probably contain too much redundant information** \
> (A10) We feel that the current Table 2 and 3 involve the minimum amount of information to study the impact of the proposed method on memory reduction for various pruning methods, $N_s$, and pruning rates. If you have any specific suggestions or directions about how Table 2 and 3 can be revised to alleviate redundancy, we would be very happy to do so.

---

### Official Review · Reviewer_dSLA · 2021-11-02

**Correctness:** 3
**Technical Novelty And Significance:** 4
**Empirical Novelty And Significance:** 4
**Recommendation:** 6
**Confidence:** 3

**Details Of Ethics Concerns:**

None.

**Main Review:**

The method is novel.  I have the following:

Concerns:

- The compression scheme with associated encoder/decoder purely solves the communication problem where HW performance is bottlenecked by limited memory bandwidth under irregular access patterns.  How does compute factor into the picture?  How does compute overhead of decoding scale?
- Experiments are performed on GPUs.  It would be desirable to understand applicability to other accelerator architectures, specifically spatial architectures.
- On page 2: ``higher sparsity leads to higher variance on pruned weights in a block".  Is this true?  Variance of the number of zero/nonzero should be nonmonotonic w.r.t. sparsity; extremely sparse or dense cases have zero variance.  Am I missing something?

**Summary Of The Paper:**

The authors proposed an encoding scheme and decoding mechanism for efficient communication of irregular sparse DNN parameters, and demonstrated with sparsified transformer and resnet50.


**Summary Of The Review:**

Novel idea of significance and potential practical value.  Presentation could use further clarity (e.g. elaboration of basic concepts such as fixed-to-fixed) since the readership of this conference is probably not experts in communications.

---

> ### Author Response · Authors · 2021-11-17
> **Response to Reviewer dSLA**
>
> We thank you for your positive feedback and effort to review the manuscript.
>
> **R1: The compression scheme purely solves the communication problem where HW performance is bottlenecked by limited memory bandwidth. How does compute overhead of decoding scale?** \
> A1: Please refer to our responses to reviewer NvPC and ieBc in terms of practical acceleration performance. Admittedly, our primary focus is to fully utilize memory bandwidth by presenting fixed-to-fixed sparsity formats. Such high utilization of memory bandwidth would be (already) highly effective to expedite inference of numerous memory-intensive models (e.g., Transformers). Then, similarly to previous approaches to alleviate computations, our proposed method can also skip zeroes (for a reduction on computations) since our sparsity formats keep masking information. We fully understand that such benefits eventually need to be demonstrated by proper hardware designs (in the form of ASICs of FPGAs). Our intention in this manuscript is to provide fundamental theories and background to enable such futuristic hardware designs (with XOR-gate decoders as memory decompressors).
>
> **R2: Experiments are performed on GPUs. It would be desirable to understand applicability to other accelerator architectures.** \
> A2: In our proposed techniques, algorithms that need to be run by GPUs include encoding procedures only. Since weight encoding can be performed offline, speed-up for encoding is not a concern for running inference. Thus, even though we illustrated our encoding algorithms using GPUs, any computing system would be fine to run weight encoding. For weight decoding, appropriate forms for hardware implementation would include ASICs and FPGAs (as we added such a discussion in the revised manuscript).
>
> **R3: On Page 2: higher sparsity leads to higher variance on pruned weights in a block. Is this true?** \
> A3: Thanks a lot for your careful review. We replaced “higher variance” with “higher standard deviation (i.e., coefficient of variation). We also added more detailed discussions in Appendix A.

---

### Official Review · Reviewer_ieBc · 2021-11-03

**Correctness:** 4
**Technical Novelty And Significance:** 2
**Empirical Novelty And Significance:** 2
**Recommendation:** 5
**Confidence:** 3

**Main Review:**

At a high level, the paper has two sides: a method and the context in which it is intended to be used. The authors do a good job at presenting the first. The technique is described in (sometimes intense) detail, and the evaluation gives straight answers to clearly-defined questions. The latter (context) is a bit weaker. The authors choose to treat their problem in isolation, which means that the paper sometimes loses sight of why the method was proposed to begin with. Computational aspects are glossed over completely, despite being the nominal motivator for the paper, and outside of a couple of statistical assumptions, there's not much connection to the ML models they test. The optimistic view is that their approach is successful regardless of input; the pessimistic view is that the impacts of various model characteristics which this community cares about are largely unknown. Appendix E does add a bit of insight here.

Strengths:

- The authors are fairly comprehensive in explaining the factors involved in choosing the method's tuning parameters. They clearly explain trade-offs and provide instructions on how to arrive at ideal values.

- I appreciated the minor analyses the authors ran. The bit inversion technique, for instance, concisely raised a problem and summarily offered a solution. Walking the reader through the pitfalls of a new technique makes the paper more enjoyable to read.

- The prose is generally clean and easy to read.

Weaknesses:

- No mention of the actual computational performance of the method. This seems important to the paper, considering the whole motivation for fixed encoding is memory access regularity. I don't necessarily expect a highly-tuned implementation that beats vendor kernels, but if the nominal goal of pursuing fixed-sized encoding is to improve speed via improved memory behavior, I would expect some concrete evidence that the proposed method achieves that purpose and some notion of the computational costs it pays for that benefit. From the method described, it looks efficient (especially if offloaded to specialized hardware), but it hurts the paper not to see quantitative support.

- I was expecting to see a mention of end-to-end model accuracy. With best the $E$ values listed, I anticipate the end-to-end effect is (hopefully) marginal, but it would be good to confirm that this is the case.
- In some cases, the paper's math could be clearer. Terms occasionally appeared in figures before their introduction in text, and I would've appreciated a brief mention of word length somewhere in section 2 or 3. It didn't help that many important terms were $N_{foo}$ or $n_{bar}$. This is minor and admittedly subjective.

Q: This work seems to share many parts with the references it seems to build on: Kwon20 and to a lesser extent Ahn19. While I have a general sense of where this paper diverges, can the authors briefly summarize what they feel the novel contributions to the community over those works are?

And as feedback: given the overlap, it might strengthen the paper to be more explicit about the differentiating factors. I would probably have appreciated a direct comparison (perhaps in a related works section, seeing as one is not supplied).

Q: Many other aggressive encoding schemes leverage the pruning mechanism to improve encoding behavior (e.g., weight sharing to reduce indexing cost for CSR schemes). These reduce encoding costs to lower than would otherwise be statistically expected. What's the analog here? I.e., with additional manipulation of non-zero weights (lossy or not), does this method offer avenues for improvement?

**Summary Of The Paper:**

The authors propose a sparse weight encoding scheme based on XOR-gate networks and a method for finding the requisite sequence which minimizes encoding error. They show that their method can achieve near-zero overhead at low encoding-error rates.

**Summary Of The Review:**

The paper does a reasonable job of presenting its sparse encoding scheme, and the analyses and evaluation seem reasonable. The paper feels a bit isolated and theoretical, which is odd given its stated goals. The method doesn't seem bad, but the reader is left with a lot of open questions about putting it into practice.

---

> ### Author Response · Authors · 2021-11-17
> **Response to Reviewer ieBc**
>
> We sincerely appreciate your careful review.
>
> **R1: No mention of the actual computational performance of the method.** \
> A1: Please refer to our responses to reviewer NvPC. Since our decoding process allows fixed-to-fixed sparsity formats, the overall performance would be mainly determined by the throughput of XOR-gate decoders. Please understand that each XOR gate requires only 6 transistors and successful circuit designs of such XOR-gate decoders have been implemented by error correction codes (for digital communication) and VLSI testing designs (with negligible area overhead for a lot of XOR logic). We acknowledge that our proposed method would be best implemented in the form of ASICs or FPGAs because performing XOR gates with CPUs or GPUs would be expensive. We expect that XOR gates can be added to the accelerator designs for DNNS in the future as well, as we observed such adoption in the area of digital communication and VLSI testing.
>
> **R2: I was expecting to see a mention of end-to-end model accuracy.** \
> A2: As described in Section 4, in this manuscript, we consider lossless compression that can be enabled by adding correction information (that is discussed in detail in Appendix F). Since E is close to 100%, the amount of correction information is tiny and our experimental results on memory save consider additional correction information in Table 2. In other words, given any pruning methods, our sparsity formats do not alter model accuracy and we can reconstruct the original weights perfectly (when combined with mask information). Thus, our focus lies in investigating encoding efficiency and memory reduction for various pruning techniques and rates.
>
> **R3: This work seems to share many parts with the references it seems to build on: Kwon20 and to a lesser extent Ahn19.** \
> A3: According to your review comments, we added Appendix C to present related works. As written in the last paragraph of Section 5, the work of Kwon20 is included and compared already in Table 2 and 3 since the results of Kwon20 is a subset of our work (as a case when $N_s=0$). The publication of Ahn19 can also be described as a special case of $N_{in}=1$. We have not discussed the case of $N_{in}=1$ in our manuscript, since $N_{in}=1$ seriously degrades throughput of decoding process (i.e., only 1 bit as input is allowed).
>
> **R4: With additional manipulation of non-zero weights (lossy or not), does this method offer avenues for improvement?** \
> A4: Our proposed method can be combined with any additional manipulation of non-zero weights providing that such manipulation does not significantly impair the assumption that pruning each weight is basically a Bernoulli trial. We feel that if the underlying pruning mechanism is performed in a fine-grained manner, any additional manipulation would not result in a structured form of pruning (e.g., row-wise or channel-wise). Correspondingly, our compression principles would not be damaged for such cases.

---

> > ### Comment · Reviewer_ieBc · 2021-11-22
> > **Further responses**
> >
> > Thank you for the rebuttal. While my general feeling towards the paper is not substantially changed, I appreciate the explanations as well as the updates to the paper. I've added additional replies below.
> >
> > ----
> >
> > *A1: Performance is tied to hardware acceleration*
> >
> > R1': If you're going to punt on CPU/GPU performance, then it feels like the paper may be focused on the wrong things. You're not actually proposing an algorithm, you're proposing a piece of hardware (and the algorithm for it, too). Since that's the case, you probably need to compare your approach to alternative sparse accelerators, and it raises more questions about trade-offs, integration, and model applicability. There's nothing wrong with proposing a method for specialized hardware, but since applicability is narrower, the bar for improvement is higher and the baseline for comparison changes.
> >
> > In addition, the paper's problem definition leans on memory behavior of conventional architectures. Since your approach is not viable for conventional hardware, this is a little bit misleading. Some characteristics (DRAM burst width, prefetch efficacy, address overhead) will almost certainly carry over to whatever accelerator implements your algorithm, but the baseline for comparison seems fundamentally different. R3' expands on this a bit, too.
> >
> > I should also mention: I'm not the only reviewer who assumed you were trying to run on conventional hardware. Most of us did. That suggests the paper may be framed incorrectly.
> >
> > *A2: Compression is lossless*
> >
> > R2': I see. It was not immediately apparent that the correction was in use for all the experiments. I incorrectly read that section as "here is a lossless option if you would rather have that".
> >
> > Since this is included, the paper should probably more accurately reflect its inclusion. For instance, Figure 6 omits it (a 'black box' for error correction would be sufficient--you needn't include the entirety of Figure S.11). However, the paper should also be a bit more truthful about the implementation. For instance, the newly-added sentence in the middle of page 6 strongly implies the solution can be implemented with a handful of transistors, which ignores lossless correction. I agree that the cost of the correction block is not high, but if you're counting it for accuracy, you should count it for cost and as part of the design.
> >
> > *A3: Added Appendix C*
> >
> > R3': I will suggest that an appendix is not a particularly suitable place for related work, especially if the method being presented is a close derivative of prior work. To your credit, you do already mention the special case of Kwon20 (very end of section 5) and Ahn19 (very beginning of section 4), but they're quite tucked away and easy to miss. I was also hoping that there was a bit more novelty in this work over those two citations that I had not caught in my reading.
> >
> > Also, given that you are proposing custom hardware, the critique of CSR on conventional platforms is perhaps a bit overstated. The reference you added to Han 2016a is perhaps the most useful, but I believe sell it short---the implementation does have memory inefficiencies, but it also represents a much fairer comparison to your method than any of the other CSR/CSC papers you cite.
> >
> > *A4: The method assumes randomized structure and values and does not perform better (or worse) if that changes.*
> >
> > That's a fair position. I don't consider this a downside to the paper. It was merely a question about possibilities.
> >
> > ----
> >
> > In general, I still feel that while the paper is technically sound in what it describes, it doesn't do a great job of justifying itself versus other SOTA hardware and it's not a substantial leap forward from the paper it builds off. If the authors still feel the work is meaningful, then I suggest it might be better if this were rewritten as a hardware paper, evaluated quantitatively against other sparse accelerators, and probably submitted to a different venue.

---

> > > ### Author Response · Authors · 2021-11-23
> > > **Thanks a lot for additional comments!**
> > >
> > > Thanks a lot for your additional replies. While we acknowledge your concerns, let us discuss the importance of our proposed work and our thoughts on why ICLR would be the best venue for our fixed-to-fixed compression technique.
> > >
> > > **R1: You’re not actually proposing an algorithm, you’re proposing a piece of hardware (and the algorithm for it, too). In addition, the paper’s problem definition leans on memory behavior of conventional architectures.** \
> > > A1: First of all, our apologies if most reviewers felt that the proposed decoding algorithm is assumed to be implemented by CPUs or GPUs. Please understand that in the revised manuscript, we clearly mentioned that designing ASICs and FPGAs are the best ways to support the proposed decoding process. We firmly believe that this work may be able to deliver a message, “while structured pruning methods have been actively studied due to inherently high parallelism (by maintaining regular memory accesses), fine-grained and unstructured pruning methods can be practical as well by avoiding irregular memory accesses patterns,” to the model compression community which would take a good amount of audience of ICLR. So far, most researchers have not suggested what would be the best DNN-dedicated sparsity representation. Instead, conventional sparsity expressions (such as CSR) have still been adopted. Hence, our work can be considered as a new DNN-oriented sparsity representation that follows the rule of “fixed-to-fixed compression format” that is useful for any kind of hardware design recognizing parallelism. We believe that the principles in Figure 1 can be applied to any kind of DNN accelerator. \
> > > Our proposed encoding/decoding algorithms can be justified because 1) various pruning methods meet the requirement of ‘almost Bernoulli event’ when pruning a weight, 2) the range of pruning rates is not too high nor too low to achieve high encoding efficiency, and 3) encoding efficiency is almost 100% such that lossless compression is obtainable. Hence, our focus in this work is verifying those justifications such that the claim “fixed-to-fixed representation is available with well-known pruning methods for various DNNs” even though special hardware designs (of XOR gates and shift registers only) can be useful. We feel that our work can expedite the research of ‘fine-grained pruning’ methods in ICLR while new fine-grained pruning methods may need to be investigated to check whether ‘fixed-to-fixed compression’ can be applied to those new pruning methods based on our proposed encoding/decoding algorithms.
> > >
> > > **R2: The newly-added sentence in the middle of page 6 strongly implies the solution can be implemented with a handful of transistors, which ignores lossless correction. You should count it for cost and as part of the design.** \
> > > A2: Thank you for your highly valuable suggestions. As you correctly indicated, Figure 6 does not include such correction parts (due to limited space and readability) while Appendix F includes details of the correction process for lossless compression. Please understand that while we describe detailed structures of XOR-gate decoders that may not be efficiently implemented by CPUs or GPUs, correction can be easily performed by simple instructions for data manipulations. Thus, we feel that our proposed correction process does not add additional cost (nor latency) to any DNN accelerators as long as such hardware supports basic data control. We will add related discussions in the final manuscript.

---

> > > > ### Author Response · Authors · 2021-11-23
> > > > **Continued..**
> > > >
> > > > **R3: I will suggest that an appendix is not a particularly suitable place for related work, especially if the method being presented is a close derivative of prior work. The critique of CSR on conventional platforms is perhaps a bit overstated.** \
> > > > A3: We would like to mention that the case of N_s=0 (which is the work of Kwon20) is thoroughly explored in Section 5 while we mentioned it at the end of Section 5. We will clearly write such information in the early part of Section 5 in the final manuscript. Accordingly, we will put more detailed comparisons with Kwon20 and Ahn19 in the final manuscript. In this work, we newly suggested encoding efficiency (which is based on entropy theory in Appendix D) as a metric measuring compression capability (such that comparison with Kwon20 becomes clear in our work). Ahn19 assumes a particular quantization method to represent weights while our work does not have such assumptions. We will add more discussion. \
> > > > We feel that Han2016a work studies hardware designs given CSR format, while we study new sparsity formats. Specifically, Han2016a mentions that the ‘load balancing’ problem issued by CSR can be somewhat mitigated but cannot be perfectly eliminated, while our proposed ‘fixed-to-fixed compression format’ inherently excludes such ‘load balancing’ issues. Thus, we believe that our work enables more fundamental and algorithmic solutions to resolve ‘irregular sparsity’ induced by fine-grained pruning that has been a roadblock to practical applications. When someone is willing to design our proposed compression methods with new models or new pruning methods, he/she would need to conduct experiments to investigate encoding efficiency and memory save (as shown in Table 2) in terms of feasibility or efficacy study. Such futuristic research would be presented in a venue similar to ICLR.
> > > >
> > > > Please let us know if you have any other concerns. We would be very happy to address.

---

### Official Review · Reviewer_NvPC · 2021-11-08

**Correctness:** 4
**Technical Novelty And Significance:** 3
**Empirical Novelty And Significance:** 3
**Recommendation:** 8
**Confidence:** 3

**Main Review:**

I like the motivation of this paper which is a very important problem in practice. To the best of my knowledge, the solution is novel and interesting, and I believe it is an orthogonal contribution to many other literatures in the compression community (e.g., magnitude-based compression, variational dropout etc). The experimental results look promising and the theoretical analysis is sound to me. I have a couple of comments regarding how the paper can be improved in the revised version. First, the paper starts with an motivation of improving the parallelism running the pruned network. However, it's a pity that the paper doesn't show any practical acceleration on ImageNet or WMT. Only compression ratio and memory reduction have been showed for various of methods. (Although Appendix A kind of showing sparse matrix multiplication is slow, I think it's still better to directly show the runtime reduction using the pruned model on real data). Second, the figures are somewhat cluttered and a bit hard to understand. I would recommend spend some time and revise them. Third, literatures and references spread across sections. An individual section (Related Work) for discussing compression literatures would be preferred.

**Summary Of The Paper:**

This paper tackles an interesting and important question which is how can we make the pruned network regular such that it can run quickly on GPU / common hardwares. The problem essentially originates from the fact that typical computational hardware can be slow when accessing in-contiguous memory address. This paper proposes a sequential encoding-decoding shemes and can compress irregular pruned weights to a regular (block-wise) structure stored in memory and shows experimental results in large-scale models / datasets.

**Summary Of The Review:**

The motivation is persuading and the method is novel. The author also demonstrate the effectiveness of their method on large scale datasets with a number of compression techniques. Although the writing can be improved, I think this is a good paper.

---

> ### Author Response · Authors · 2021-11-17
> **Response to Reviewer NvPC**
>
> We greatly appreciate your positive feedback and comments.
>
> **R1: The paper doesn’t show any practical acceleration on ImageNet or WMT.** \
> A1: We acknowledge that demonstrating practical performance in terms of latency reduction would be the best to present the unique strength and benefits of the proposed method. Unfortunately, however, such overall performance at the system level would be highly dependent on the overall hardware design and specific architectural choices (while we focus on the fundamental theories in the aspect of parallelism in this manuscript). As discussed in the manuscript, designing ASICs or FPGAs (especially to implement XOR-gate decoders to fully enable the parallel weight decoding process) would be highly effective to see such practical acceleration. We believe that our work is the first practical step to envision fixed-to-fixed sparsity formats such that DNN accelerator designers can consider fine-grained pruning as an efficient and feasible acceleration method. Please understand that we felt that it is necessary to spend most space of the manuscript to suggest some theoretical background and empirical evidence that random number generators can provide high encoding efficiency for various pruning techniques. We expect that active hardware research can be encouraged in the future by our proposed method presenting fixed-to-fixed sparsity format and lossless compression.
>
> **R2: The figures are somewhat cluttered and a bit hard to understand** \
> A2: We agree with you that some figures are busy and some readers may take some time to fully understand the figures in the manuscript. Figure 6 has been described in more detail in the revised manuscript. Please let us know which figures are specifically cluttered. We will try our best to improve the quality of the figures as long as the space is allowed.
>
> **R3: Literatures and references are spread across sections. An individual section (Related Work) for discussing compression literature would be preferred.** \
> A3: Thank you so much for your valuable suggestion. We added Appendix C to include related works.

---

### Author Response · Authors · 2021-11-17
**Revised manuscript is uploaded**

We thank all the reviewers for their valuable comments and detailed feedback. Please find an uploaded revised manuscript. We revised the manuscript according to the reviewers' comments to address concerns and to avoid any unnecessary confusion. All changes in the revised manuscript are highlighted in red.


- We added Appendix A to explain why memory bandwidth is wasted for fixed-to-variable sparsity format as shown in Figure 1 (Section 2 has been revised accordingly).
- Related works are presented in Appendix C.
- Figure 6 is described in the caption with more details.
- We replaced “XOR-gate network” with “XOR-gate decoder.”

---

### Comment · Reviewer_Qcma · 2021-11-17
**Please show the space and time complexity of the encoding and decoding process**

Dear authors,

I understand that demonstrating the actual performance advantage would be difficult. Though please try your best to estimate the time and space complexity of your algorithm. So we can have a better idea about the practicality of your approach.

---

> ### Author Response · Authors · 2021-11-17
> **Our response on the complexity analysis**
>
> Thank you so much for your prompt response. Let us discuss issues on the design complexity of our proposed compression method that can be divided into encoding and decoding steps. First, as we addressed below, the time and space complexity of **encoding** process is not a concern since encoding is performed offline and independent of inference performance (thus, GPUs or CPUs would be fine to run encoding algorithm even though we choose GPUs in the manuscript along with an exemplary encoding algorithm based on a dynamic programming technique). Second, we suggest that the decoding algorithm is supported by a hardware design consisting of XOR gates (and a few shift registers). Hence, in an effort to address your concerns, we feel that it is necessary to argue hardware design issues of XOR-gate decoders.
>
> - The strongest benefit of employing digital circuits (in the form of ASICs or FPGAs) to implement XOR gates is that all XOR gates can be performed simultaneously (unlike GPUs or CPUs where each core needs to simulate only a few XOR gates). Thus, **all XOR operations of our proposed decoder are completed within just one clock cycle.**
>
> -  $N_s$ (with shift registers) would increase the latency. Throughput, however, maintains to be the same regardless of $N_s$ through pipelining technique which is a basic hardware design principle.
>
> - Overall, the design complexity (in terms of area overhead and latency) of XOR-gate decoders is extremely low (note that one XOR gate consumes only 6 transistors).
>
> - **XOR-gate decoders would work as memory decompressors (located in-between memory and computation logic).** In the view of computational units that receive outputs of an XOR-gate decoder, then, the amount of memory is simply reduced while regular memory access patterns are not disturbed.
>
> - **In the case of $N_s =2$, the overall design cost would be 2 additional clock cycles for the latency and a few hundred XOR gates (which would be approximately equivalent to a few thousand transistors).** Overall, we can provide full memory bandwidth (based on regular memory access patterns through fixed-to-fixed sparsity formats) while the overall hardware design cost is only marginal.
>
> - While designing DNN inference accelerators is gaining increasing attention, our work can provide a new research direction.
>
> Please let us know if you have any additional questions or concerns on the implementation cost of the proposed method.

---

> > ### Comment · Reviewer_Qcma · 2021-11-18
> > **Please elaborate on the derivation of complexity calculation**
> >
> > Dear authors,
> >
> > Please elaborate on the derivation process of (time and space) complexity results, given this is the major concern of almost all reviewers. And I suggest including this as a part of your manuscript.

---

> > > ### Author Response · Authors · 2021-11-19
> > > **We added the complexity analysis in the manuscript**
> > >
> > > Thank you so much for your valuable suggestion. We revised the manuscript to add Appendix G that includes the complexity analysis we discussed throughout our responses to the reviewers. Correspondingly, we added one sentence in the last part of Section 4 to indicate Appendix G.
> > >
> > > We would be grateful if you have any more particular suggestions to improve the manuscript.

---

> > > > ### Comment · Reviewer_Qcma · 2021-11-20
> > > > **Appendix G**
> > > >
> > > > Dear authors,
> > > >
> > > > I was expecting to see some equations that one can use to calculate the complexity numbers. "A few hundred" "A few thousand" are too vague for a research paper like this. And there is no mention of the memory complexity and complexity for the encoding process.

---

> > > > > ### Author Response · Authors · 2021-11-20
> > > > > **Appendix G is revised**
> > > > >
> > > > > We apologize that Appendix G was not clearly explaining encoding and decoding complexity.
> > > > >
> > > > > We revised Appendix G thoroughly to include detailed equations and analysis for encoding process and decoding process.
> > > > >
> > > > > Please let us know if you have any other comments or suggestions to improve the clarity of the manuscript.
> > > > > Thank you so much for your prompt responses.

---

### Decision · Program_Chairs · 2022-01-20

**Decision:**

Accept (Poster)

**Comment:**

### Summary

The key idea behind this approach is a new technique to map irregular sparsity to a regular, compressed pattern. The results can, in principle, therefore overcome several standard limitations with irregular data storage formats.  The results improve over existing (though related) techniques.

### Discussion

#### Strenghts

- An interesting and timely topic to study

- Results show non-compute improvements

#### Weakness

The primary weakness noted among the reviewers was the lack of study on actual decoding performance. As I note below, this is a serious oversight that given the already existing theoretical work in the area warrants study as the community should begin to turn towards mapping that theory to practice.

### Recommendation

I recommend Accept (poster). This is a strong piece of theoretical work. However,  I would like to note that while I believe this work meets the current evaluation standards set in the area, it is time for follow on work to take the additional step to validate the practicality of the approach through a performance evaluation (either in simulation or FPGA/ASIC work).